# WHITENING AND COLORING BATCH TRANSFORM FOR GANS

**Aliaksandr Siarohin, Enver Sangineto & Nicu Sebe**
Department of Information Engineering and Computer Science (DISI)
University of Trento, Italy
{aliaksadr.siarohin,enver.sangineto,niculae.sebe}@unitn.it

## ABSTRACT

Batch Normalization (BN) is a common technique used to speed-up and stabilize training. On the other hand, the learnable parameters of BN are commonly used in conditional Generative Adversarial Networks (cGANs) for representing class-specific information using conditional Batch Normalization (cBN). In this paper we propose to generalize both BN and cBN using a Whitening and Coloring based batch normalization. We show that our conditional Coloring can represent categorical conditioning information which largely helps the cGAN qualitative results. Moreover, we show that full-feature whitening is important in a general GAN scenario in which the training process is known to be highly unstable. We test our approach on different datasets and using different GAN networks and training protocols, showing a consistent improvement in all the tested frameworks. Our CIFAR-10 conditioned results are higher than all previous works on this dataset.

## 1 INTRODUCTION

Generative Adversarial Networks (GANs) (Goodfellow et al. (2014)) have drawn much attention in the last years due to their proven ability to generate sufficiently realistic short videos (Vondrick et al. (2016)) or still images (Gulrajani et al. (2017)), possibly *conditioned* on some input information such as: a class label (Odena et al. (2017)), an image (Isola et al. (2017); Tang et al. (2019)), a textual description (Reed et al. (2016)), some structured data (Tang et al. (2018); Siarohin et al. (2018a)) or some perceptual image attribute (Siarohin et al. (2018b))). When the generation process depends on some input data, the framework is commonly called conditional GAN (cGAN).

In this paper we deal with both conditional and unconditional GANs and we propose to replace Batch Normalization (Ioffe & Szegedy (2015)) (BN) with a *Whitening and Coloring* (WC) transform (Hossain (2016)) in order to jointly speed-up training and increase the network representation capacity. BN, proposed in (Ioffe & Szegedy (2015)) for discriminative tasks and then adopted in many discriminative and generative networks, is based on feature *standardization* of all the network layers. The original motivation behind the introduction of BN is to alleviate the internal Covariate Shift problem caused by the continuous modification of each layer's input representation during training (Ioffe & Szegedy (2015)). However, recent works (Santurkar et al. (2018); Kohler et al. (2018)) showed that a major motivation of the empirical success of BN relies also on a higher training stability, the latter being due to a better-conditioning of the Jacobian of the corresponding loss when BN is used compared to training without BN. Basically, feature normalization makes the loss landscape smoother, hence it stabilizes and speeds-up training. Based on this intuition, BN has very recently been extended in (Huang et al. (2018)) to full-feature decorrelation using ZCA whitening (Kessy et al. (2017)) in a discriminative scenario.

On the other hand, GAN training is known to be particularly unstable, because GAN optimization aims at finding a Nash equilibrium between two players, a problem which is more difficult and less stable than the common discriminative-network optimization and that frequently leads to non-convergence. Specifically, Odena et al. (2018) show the relation between stability of GAN training and conditioning of the Jacobian. This motivates our proposal to extend BN to full-feature whitening in a GAN scenario (more details on this in Sec. 6). Differently from Huang et al. (2018), our feature whitening is based on the Cholesky decomposition (Dereniowski & Marek (2004)), and we

empirically show that our method is much faster and stable and achieves better results in GAN training with respect to ZCA whitening. Importantly, while in (Huang et al. (2018)) feature whitening is followed by a *per-dimension scaling and shifting* of each feature, as in the original BN (Ioffe & Szegedy (2015)), we propose to use a *coloring* transform (Hossain (2016)) to keep the network representation capacity unchanged with respect to a non-whitened layer. Our coloring projects each whitened feature vector in a new distribution using learnable *multivariate* filters and we show that this transformation is critical when using whitening for GANs.

The second contribution of this paper is based on the aforementioned coloring transform to represent class-specific information in a cGAN scenario. Formally, given a set of class labels $\mathcal{Y} = \{y_1, ..., y_n\}$, we want to generate an image $I = G(\mathbf{z}, y)$, where the generator $G$ is input with a noise vector $\mathbf{z}$ *and* a class label $y$. For instance, if $y = cat$, we want a foreground object in $I$ depicting a cat, if $y = dog$, then $I$ should represent a dog, etc. The categorical conditional information $y$ is commonly represented in $G$ using *conditional batch normalization* (cBN) (Dumoulin et al. (2016b); Gulrajani et al. (2017); Miyato & Koyama (2018)), where a *class-specific* pair of *scaling-shifting* parameters is learned for each class $y$. In our proposed *conditional Whitening and Coloring* (cWC) the class-specific scaling-shifting parameters are replaced with a combination of class-agnostic and class-specific coloring filters. The intuitive idea behind the proposed cWC is that multivariate transformation filters are more informative than the scalar parameters proposed in (Dumoulin et al. (2016b)) and, thus, they can more accurately represent class-specific information.

Finally, in order to have a set of class-specific coloring filters ($\{\Gamma_y\}$) which grows sub-linearly with respect to the number of classes $n$, we represent each $\Gamma_y$ using a common, restricted dictionary of class-independent filters together with class-filter soft-assignment weights. The use of a class-independent dictionary of filters, together with a class-agnostic branch in the architecture of our cGAN generators (see Sec. 4) makes it possible to share parameters over all the classes and to reduce the final number of weights which need to be learned.

In summary, our contributions are the following:

- We propose a whitening-coloring transform for batch normalization to improve stability of GAN training which is based on the Cholesky decomposition and a re-projection of the whitened features in a learned distribution.

- In a cGAN framework we propose: (1) A class-specific coloring transform which generalizes cBN (Dumoulin et al. (2016b)) and (2) A soft-assignment of classes to filters to reduce the network complexity.

- Using different datasets and basic GAN-cGAN frameworks, we show that our WC and cWC consistently improve the original frameworks' results and training speed.

Our code is publicly available `https://github.com/AliaksandrSiarohin/wc-gan`.

## 2 RELATED WORK

**Batch Normalization.** BN (Ioffe & Szegedy (2015)) is based on a batch-based (per-dimension) standardization of all the network's layers. Formally, given a batch of $d$-dimensional samples $B = \{\mathbf{x}_1, ..., \mathbf{x}_m\}$, input to a given network layer, BN transforms each sample $\mathbf{x}_i \in \mathbb{R}^d$ in $B$ using:

$$BN(x_{i,k}) = \gamma_k \frac{x_{i,k} - \mu_{B,k}}{\sqrt{\sigma_{B,k}^2 + \epsilon}} + \beta_k, \tag{1}$$

where: $k$ ($1 \leq k \leq d$) indicates the $k$-th dimension of the data, $\mu_{B,k}$ and $\sigma_{B,k}$ are, respectively, the mean and the standard deviation computed with respect to the $k$-th dimension of the samples in $B$ and $\epsilon$ is a constant used to prevent numerical instability. Finally, $\gamma_k$ and $\beta_k$ are *learnable* scaling and shifting parameters, which are introduced in (Ioffe & Szegedy (2015)) in order to prevent losing the original representation capacity of the network. More in detail, $\gamma_k$ and $\beta_k$ are used to guarantee that the BN transform can be inverted, if necessary, in order to represent the identity transform (Ioffe & Szegedy (2015)). BN is commonly used in both discriminative and generative networks and has been extended in many directions. For instance, Ba et al. (2016) propose Layer Normalization,

where normalization is performed using all the layer's activation values of a single sample. Luo (2017) proposes to whiten the activation values using a *learnable* whitening matrix. Differently from (Luo (2017)), our whitening matrix only depends on the specific batch of data. Similarly to our work, batch-dependent full-feature decorrelation is performed also by Huang et al. (2018), who use a ZCA-whitening and show promising results in a discriminative scenario. However, the *whitening* part of our method is based on the Cholesky decomposition and we empirically show that, in a GAN scenario where training is known to be highly unstable, our method produces better results and it is much faster to compute. Moreover, while in (Huang et al. (2018)) only scaling and shifting parameters are used after feature decorrelation, we propose to use a *coloring* transform. Analogously to the role played by $\gamma_k$ and $\beta_k$ in Eq. 1, coloring can potentially invert the whitening transform if that were the optimal thing to do, in this way preserving the original representation capacity of the network . We empirically show (see Sec. 5.1.1) that whitening without coloring performs poorly. Last but not least, our *conditional coloring* can be used to represent rich class-specific information in a cGAN framework (see below).

**Representing conditional information in cGANs.** cGANs have been widely used to produce images conditioned on some input. For instance, Isola et al. (2017) transform a given image in a second image represented in another "channel". Siarohin et al. (2018a) extend this framework to deal with structured conditional information (the human body pose). In (Reed et al. (2016)) a textual description of an image is used to condition the image generation. When the conditioning data $y$ is a categorical, unstructured variable, as in the case investigated in this paper, information about $y$ can be represented in different ways. cBN (Dumoulin et al. (2016b)) uses $y$ to select class-specific scaling and shifting parameters in the generator (more details in Sec. 4). Another common approach is to represent $y$ using a one-hot vector, which is concatenated with the input layer of both the discriminator and the generator (Mirza & Osindero (2014); Perarnau et al. (2016)) or with the first convolutional layer of the discriminator (Perarnau et al. (2016)). Miyato & Koyama (2018) split the last discriminator layer in two branches. The first branch estimates the class-agnostic probability that the input image is real, while the second branch represents class-conditional information. This solution is orthogonal to our proposed cWC and we show in Sec. 5.2 that the two approaches can be combined for boosting the results obtained in (Miyato & Koyama (2018)).

## 3 THE WHITENING AND COLORING TRANSFORM

We start describing the details of our Whitening and Coloring transform in the unconditional case, which we refer to as WC in the rest of the paper. We always assume an image domain and convolutional generator/discriminator networks. However, most of the solutions we propose can potentially be applied to non-visual domains and non-convolutional networks.

Let $F \in \mathbb{R}^{h \times w \times d}$ be the tensor representing the activation values of the convolutional feature maps for a given image and layer, with $d$ channels and $h \times w$ spatial locations. Similarly to BN, we consider the $d$-dimensional activations of each location in the $h \times w$ convolutional grid as a separate instance $\mathbf{x}_i \in \mathbb{R}^d$ in $B$. This is done also to have a larger cardinality batch which alleviates instability issues when computing the batch-related statistics. In fact, a mini-batch of $m'$ images corresponds to $m = |B| = m' \times h \times w$ (Ioffe & Szegedy (2015)).

Using a vectorial notation, our proposed WC-based batch normalization is given by $WC(\mathbf{x}_i) = Coloring(\hat{\mathbf{x}}_i)$, where $\hat{\mathbf{x}}_i = Whitening(\mathbf{x}_i)$:

$$Coloring(\hat{\mathbf{x}}_i) = \Gamma\hat{\mathbf{x}}_i + \boldsymbol{\beta} \qquad (2)$$

$$Whitening(\mathbf{x}_i) = W_B(\mathbf{x}_i - \boldsymbol{\mu}_B). \qquad (3)$$

In Eq. 3, the vector $\boldsymbol{\mu}_B$ is the mean of the elements in $B$ (being $\mu_{B,k}$ in Eq. 1 its $k$-th component), while the matrix $W_B$ is such that: $W_B^\top W_B = \Sigma_B^{-1}$, where $\Sigma_B$ is the covariance matrix computed using $B$. This transformation performs the full *whitening* of $\mathbf{x}_i$ and the resulting set of vectors $\hat{B} = \{\hat{\mathbf{x}}_1, ..., \hat{\mathbf{x}}_m\}$ lies in a spherical distribution (i.e., with a covariance matrix equal to the identity matrix). Note that, similarly to $\mu_{B,k}$ and $\sigma_{B,k}$ in Eq. 1, $\boldsymbol{\mu}_B$ and $W_B$ are completely data-dependent, being computed using only $B$ and without any learnable parameter involved. On the other hand, Eq. 2 performs the *coloring* transform (Hossain (2016)), which projects the elements in $\hat{B}$ onto a

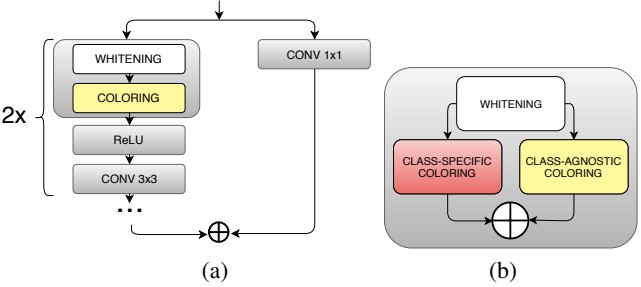

Figure 1: (a) A ResNet block composed of the sequence: *Whitening, Coloring, ReLU, $Conv_{3\times3}$, Whitening, Coloring, ReLU, $Conv_{3\times3}$.* (b) A cWC layer.

multivariate Gaussian distribution with an arbitrary covariance matrix (see Sec. 2 for the motivation behind this projection). Eq. 2 is based on the *learnable* parameters $\boldsymbol{\beta} = (\beta_1, ..., \beta_k, ..., \beta_d)^\top$ and $\Gamma = \begin{pmatrix} \cdots \\ c_k \\ \cdots \end{pmatrix}$, where $\boldsymbol{c}_k$ is a *vector* of parameters (a *coloring filter*) which generalizes the scalar parameter $\gamma_k$ in Eq. 1. Coloring is a linear operation and can be simply implemented using $1 \times 1 \times d$ convolutional filters $\boldsymbol{c}_k$ and the corresponding biases $\beta_k$. We compute $W_B$ using the Cholesky decomposition, which has the important properties to be efficient and well conditioned. We provide the details in Sec. A, where we also analyze the computational costs, while in Sec. E we compare our approach with the ZCA-based whitening proposed by Huang et al. (2018).

WC is plugged before each convolutional layer of our generators ($G$). In most of our experiments, we use a ResNet-like architecture (He et al. (2016)) for $G$ with 3 blocks (Gulrajani et al. (2017)) and a final convolutional layer (i.e. $3 \times 2 + 1 = 7$ convolutional layers, corresponding to 7 WC layers). Fig. 1(a) shows a simplified block scheme. We do not use WC nor we introduce any novelty in the discriminators (more details in Sec. 5).

## 4 CONDITIONAL COLORING TRANSFORM

In a cGAN framework, the conditional information $y$, input to $G$, is commonly represented using cBN (Dumoulin et al. (2016b)), where Eq. 1 is replaced with:

$$cBN(x_{i,k}, y) = \gamma_{y,k} \frac{x_{i,k} - \mu_{B,k}}{\sqrt{\sigma_{B,k}^2 + \epsilon}} + \beta_{y,k}. \tag{4}$$

Comparing Eq. 1 with Eq. 4, the only difference lies in the class-specific $\gamma$ and $\beta$ parameters, which are used to project each standardized feature into a class-specific univariate Gaussian distribution. We propose to replace Eq. 4 with our *conditional Whitening and Coloring* transform (cWC): $cWC(\mathbf{x}_i, y) = CondColoring(\hat{\mathbf{x}}_i, y)$ and $\hat{\mathbf{x}}_i = Whitening(\mathbf{x}_i)$, where:

$$CondColoring(\hat{\mathbf{x}}_i, y) = \Gamma_y \hat{\mathbf{x}}_i + \boldsymbol{\beta}_y + \Gamma \hat{\mathbf{x}}_i + \boldsymbol{\beta}. \tag{5}$$

The first term in Eq. 5 is based on class-specific learnable parameters $\Gamma_y$ and $\boldsymbol{\beta}_y$. Analogously to (Dumoulin et al. (2016b)), in the forward-pass (both at training and at inference time), conditional information $y$, input to $G$, is used to select the correct $(\Gamma_y, \boldsymbol{\beta}_y)$ pair to apply. Similarly, during training, gradient information is backpropagated through *only* the corresponding pair used in the forward pass. $Whitening(\mathbf{x}_i)$ is the same procedure used in the unconditional case, where *all* the elements in $B$ (independently of their class label) are used to compute $\boldsymbol{\mu}_B$ and $\Sigma_B$.

The second term in Eq. 5 is based on class-agnostic learnable parameters $\Gamma$ and $\boldsymbol{\beta}$, similarly to the unconditional coloring transform in Eq. 2. This second term is used because the class-specific $(\Gamma_y, \boldsymbol{\beta}_y)$ parameters are trained with less data then the corresponding class-agnostic weights. In fact, in a generative task with $n$ classes and a training protocol with batches of $m$ elements, on average only $\frac{m}{n}$ instances in $B$ are associated with label $y$. Thus, on average, at each training iteration, the

parameters $(\Gamma_y, \boldsymbol{\beta}_y)$ are updated using an *effective batch* of $\frac{m}{n}$ elements versus a batch of $m$ elements used for all the other class-independent parameters. The class-agnostic term is then used to enforce sharing of the parameters of the overall coloring transform. In Sec. 5.2.1 we empirically show the importance of this second term. Fig. 1(b) shows the two separate network branches corresponding to the two terms in Eq. 5.

## 4.1 CLASS-FILTER SOFT ASSIGNMENT

When the number of classes $n$ is small and the number of real training images is relatively large, learning a separate $\Gamma_y$ for each $y$ is not a problem. However, with a large $n$ and/or few training data, this may lead to an over-complex generator network which can be hard to train. Indeed, the dimensionality of each $\Gamma_y$ is $d \times d$ and learning $nd^2$ parameters may be difficult with a large $n$.

In order to solve this issue, when $n$ is large, we compute $\Gamma_y$ using a weighted sum over the elements of a class-independent dictionary $D$. Each row of $D$ contains a flattened version of a $\Gamma$ matrix. More formally, $D$ is an $s \times d^2$ matrix, with $s << n$. The $j$-th row in $D$ is given by the concatenation of $d$ $d$-dimensional filters: $D_j = [\boldsymbol{c}_1^j, ..., \boldsymbol{c}_k^j, ...\boldsymbol{c}_d^j]$, being $\boldsymbol{c}_k^j \in \mathbb{R}^d$ a coloring filter for the $k$-th output dimension. $D$ is shared over all the classes. Given $y$ as input, $\Gamma_y$ is computed using:

$$\Gamma_y = \mathbf{y}^\top A D, \tag{6}$$

where $\mathbf{y}$ is a one-hot representation of class $y$ (i.e., a vector of zero elements except one in the $y$-th component) and it is used to select the $y$-th row ($A_y$) of the $n \times s$ association matrix $A$. During training, given $y$, only the weights in $A_y$ are updated (i.e., all the other rows $A_{y'}$, with $y' \neq y$ are not used and do not change). However, *all* the elements in $D$ are updated independently of $y$.

In the rest of the paper, we call (plain) *cWC* the version in which $n$ different class-specific matrices $\Gamma_y$ are learned, one per class, and $cWC_{sa}$ the version in which $\Gamma_y$ is computed using Eq. 6. In all our experiments (Sec. 5.2), we fix $s = \lceil \sqrt{n} \rceil$.

## 5 EXPERIMENTS

We split our experiments in two main scenarios: unconditional and conditional image generation. In each scenario we first compare our approach with the state of the art and then we analyze the impact of each element of our proposal. We use 5 datasets: CIFAR-10, CIFAR-100 (Torralba et al. (2008)), STL-10 (Coates et al. (2011)), ImageNet (Deng et al. (2009)) and Tiny ImageNet[1]. The latter is composed of $64 \times 64$ images taken from 200 ImageNet categories (500 images per class). In our evaluation we use the two most common metrics for image generation tasks: Inception Score (IS) (Salimans et al. (2016)) (the higher the better) and the Fréchet Inception Distance (FID) (Heusel et al. (2017)) (the lower the better). In the Appendix we also show: (1) a comparison based on human judgments as an additional evaluation criterion, (2) qualitative results for all the datasets used in this section, (3) additional experiments on other smaller datasets.

In our experiments we use different generator and discriminator architectures in combination with different discriminator training protocols and we plug our WC/cWC in the generator of each adopted framework. The goal is to show that our whitening-coloring transforms can consistently improve the quality of the results with respect to different basic GAN networks, training protocols and image generation tasks. Note that, if not otherwise explicitly mentioned, we usually adopt the same hyperparameter setting of the framewoks we compare with. Altough better results can be achieved using, for instance, an ad-hoc learning rate policy (see Sec. 5.2 and Tab. 2 (right)), we want to emphasize that our whitening-coloring transforms can be easily plugged into existing systems.

In all our experiments, $WC$, $cWC$ and $cWC_{sa}$, as well as the simplified baselines presented in the ablation studies, are applied before *each* convolutional layer of our generators.

---

[1] https://tiny-imagenet.herokuapp.com/

### 5.1 UNCONDITIONAL IMAGE GENERATION

In the unconditional scenario we use two different training strategies for the *discriminator*: (1) WGAN with Gradient Penalty (Gulrajani et al. (2017)) (*GP*) and (2) GAN with Spectral Normalization (Miyato et al. (2018)) (*SN*). Remind that our WC is used only in the generator (Sec. 3). When WC is used in combination with *GP*, we call it *WC GP*, while, in combination with *SN*, we refer to it as *WC SN*. Moreover, we use either ResNet (He et al. (2016); Gulrajani et al. (2017)) or DCGAN (Radford et al. (2015)) as the basic generator architecture. In order to make the comparison with the corresponding *GP* and *SN* frameworks as fair as possible and to show that the quantitative result improvements are due *only* to our batch normalization approach, we strictly follow the architectural and training protocol details reported in (Gulrajani et al. (2017); Miyato et al. (2018)). For instance, in *WC SN + DCGAN* we use the same DCGAN-based generator and discriminator used in *SN + DCGAN* (Miyato et al. (2018)), with the same number of feature-map dimension ($d$) in each layer in both the generator and the discriminator, the same learning rate policy (which was optimized for *SN + DCGAN* but not for our framework), etc. We provide implementation details in Sec. C.1.

In Tab. 1 we show that WC improves both the IS and the FID values of all the tested frameworks in both the CIFAR-10 and the STL-10 dataset. Specifically, independently of the basic architecture (either ResNet or DCGAN), our WC transform improves the corresponding *SN* results. Similarly, our *WC GP* results are better than the corresponding *GP* values.

Table 1: CIFAR-10 and STL-10 results of different frameworks with and without our WC. The *GP* and *SN* based values (without WC) are taken from the corresponding articles (Gulrajani et al. (2017); Miyato et al. (2018)). For completeness, we also report *FID 10k* results following the FID-computation best practice suggested in (Heusel et al. (2017)).

| | CIFAR-10 | | | STL-10 | | |
|---|---|---|---|---|---|---|
| Method | IS | FID 5k | FID 10k | IS | FID 5k | FID 10K |
| GP + ResNet (Gulrajani et al. (2017)) | $7.86 \pm .07$ | - | - | - | - | - |
| WC GP + ResNet (ours) | $8.20 \pm .08$ | - | 20.4 | - | - | - |
| SN + DCGAN (Miyato et al. (2018)) | $7.58 \pm .12$ | 25.5 | - | $8.79 \pm .14$ | 43.2 | - |
| WC SN + DCGAN (ours) | $7.84 \pm .10$ | 25.5 | 23.0 | $9.45 \pm 0.18$ | 40.1 | 37.9 |
| SN + ResNet (Miyato et al. (2018)) | $8.22 \pm .05$ | 21.7 | - | $9.10 \pm .04$ | 40.1 | - |
| WC SN + ResNet (ours) | $8.66 \pm .11$ | 20.2 | 17.2 | $9.93 \pm .13$ | 38.7 | 36.4 |

In Tab. 2 (left) we compare *WC SN + ResNet* with other methods on CIFAR-10 and we show that we reach results almost on par with the state of the art on this dataset (Karras et al. (2018)). Note that the *average* IS values over 10 runs reported in Karras et al. (2018) is exactly the same as our *average* IS value: 8.56 (the results reported in the table refer to the best values for each framework). Moreover, Karras et al. (2018) use a much higher capacity generator (about $\times 4$ the number of parameters of our *WC SN + ResNet*) and discriminator (about $\times 18$ parameters than ours).

Finally, in Fig. 2 we plot the IS and the FID values computed at different mini-batch training iterations. These plots empirically show that the proposed batch normalization approach significantly speeds-up the training process. For instance, the IS value of *WC SN + ResNet* after 20k iterations is already higher than the IS value of *SN + ResNet* after 50k iterations.

Table 2: CIFAR-10 experiments using unconditioned (left) or conditioned (right) methods. Most of the reported results are taken from (Gulrajani et al. (2017)).

| Method | IS | | Method | IS |
|---|---|---|---|---|
| ALI Dumoulin et al. (2016a) | $5.34 \pm .05$ | | SteinGAN Wang & Liu (2016) | 6.35 |
| BEGAN Berthelot et al. (2017) | 5.62 | | DCGAN with labels Wang & Liu (2016) | 6.58 |
| DCGAN Radford et al. (2015) | $6.16 \pm .07$ | | Improved GAN Salimans et al. (2016) | $8.09 \pm .07$ |
| Improved GAN (-L+HA) Salimans et al. (2016) | $6.86 \pm .06$ | | AC-GAN Odena et al. (2017) | $8.25 \pm .07$ |
| EGAN-Ent-VI Dai et al. (2017) | $7.07 \pm .10$ | | SGAN-no-joint Huang et al. (2017) | $8.37 \pm .08$ |
| DFM Warde-Farley & Bengio (2017) | $7.72 \pm .13$ | | WGAN-GP ResNet Gulrajani et al. (2017) | $8.42 \pm .10$ |
| WGAN-GP Gulrajani et al. (2017) | $7.86 \pm .07$ | | SGAN Huang et al. (2017) | $8.59 \pm .12$ |
| CT-GAN Wei et al. (2018) | $8.12 \pm .12$ | | SNGAN-PROJECITVE Miyato & Koyama (2018) | 8.62 |
| SN-GAN Miyato et al. (2018) | $8.22 \pm .05$ | | CT-GAN Wei et al. (2018) | $8.81 \pm .13$ |
| OT-GAN Salimans et al. (2018) | $8.46 \pm .12$ | | AM-GAN Zhou et al. (2018) | $8.91 \pm .11$ |
| **PROGRESSIVE** Karras et al. (2018) | $\mathbf{8.80} \pm .13$ | | **cWC SN + Proj. Discr.** (ours) | $\mathbf{9.06} \pm .13$ |
| WC SN + ResNet (ours) | $8.66 \pm .11$ | | $cWC_{sa}$ SN + Proj. Discr. (ours) | $8.85 \pm .10$ |

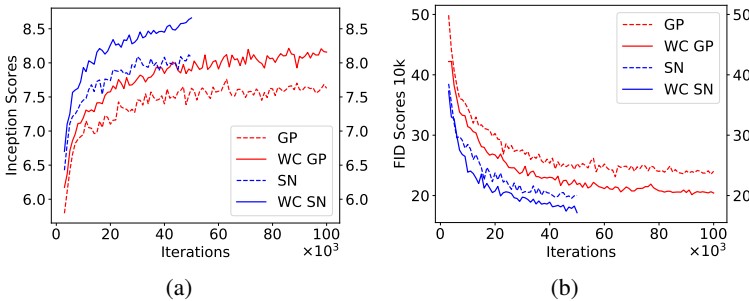

Figure 2: IS (a) and FID 10k (b) values on CIFAR-10 sampled at every 1000 iterations.

### 5.1.1 ABLATION STUDY

We use CIFAR-10 to analyze the influence of each WC element and an SN + ResNet framework (Miyato et al. (2018)), omitting 'SN' and 'ResNet' from the names for simplicity.

Our initial baseline is *W-only*, in which we use Eq. 3 to project the features in a spherical distribution *without coloring*. In *WC-diag*, after whitening (Eq. 3), we use Eq. 2 with *diagonal* $\Gamma$ matrices. This corresponds to learning *scalar* $\gamma$-$\beta$ parameters for a dimension-wise scaling-shifting transform as in BN (Eq. 1). On the other hand, in *std-C* we *standardize* the features as in BN (Eq. 1) but replacing the $\gamma$-$\beta$ parameters with our coloring transform (Eq. 2) and in *C-only*, Eq. 2 is applied directly to the original, non-normalized batch instances. Finally, *WC* is our full-pipeline.

Tab. 3 shows that feature whitening without coloring performs poorly. After the feature normalization step, it is crucial for the network to re-project the features in a new (learned) distribution with a sufficient representation capacity. This is done in standard BN using the $\gamma$-$\beta$ parameters. Analogously, in our WC, this role is played by the coloring step. In fact, with a diagonal-matrix based coloring, *WC-diag* improves the results with respect to *W-only*. Moreover, a drastic improvement is obtained by the full-pipeline (*WC*) with respect to *WC-diag*, showing the importance of learning a full covariance matrix in the coloring step. On the other hand, the comparison between *WC* with both *C-only* and *std-C* shows that feature normalization before coloring is also important, and that the full advantage of coloring is obtained when used in combination with whitening.

In the same table we report the results obtained by replacing our Cholesky decomposition with the ZCA-based whitening ($W_{zca}C$), which is used in (Huang et al. (2018)), keeping all the rest of our method unchanged (coloring included). The comparison with WC shows that our proposed procedure (detailed in Sec. A) achieves better results than $W_{zca}C$. The results reported in Tab. 3 correspond to the *best* values achieved by $W_{zca}C$ during training and in Sec. E we show that $W_{zca}C$ can be highly unstable. Moreover, $W_{zca}C$ is, on average, much slower than WC. Specifically, the whitening only part, when computed using the procedure proposed in Sec. A, is more than *11 times faster* than the ZCA-based whitening, while, considering the whole pipeline (most of which is shared between WC and $W_{zca}C$), WC is more than *3.5 times faster* than $W_{zca}C$. However, note that in the Decorrelated Batch Normalization (DBN) (Huang et al. (2018)), no coloring is used and the ZCA-based whitening is applied to only the first layer of a classification network, trained in a standard supervised fashion. Hence, $W_{zca}C$ has to be considered as our proposed WC in which whitening is performed using the procedure proposed in (Huang et al. (2018)).

Table 3: CIFAR-10: Ablation study of WC.

| Method | IS | FID 10k |
|---|---|---|
| *W-only* | $6.63 \pm .07$ | 36.8 |
| *WC-diag* | $7.00 \pm .09$ | 34.1 |
| *C-only* | $7.86 \pm .08$ | 23.17 |
| *std-C* | $8.34 \pm .08$ | 18.02 |
| $W_{zca}C$ | $8.29 \pm .08$ | 18.57 |
| *WC* | $8.66 \pm .11$ | 17.2 |

Table 4: Tiny ImageNet results.

| Method | IS | FID 10k |
|---|---|---|
| *cBN* | $9.27 \pm .14$ | 39.9 |
| *cWC* | $10.43 \pm .13$ | 35.7 |
| $cWC_{sa}$ | $11.78 \pm .25$ | 27.7 |

## 5.2 CONDITIONAL IMAGE GENERATION

In the cGAN scenario, conditional information is represented in the generator as explained in Sec. 4 and using either *cWC* or $cWC_{sa}$ (Sec. 4.1). The purpose of this section is to show the advantage of using our conditional coloring with respect to other cGAN methods for representing categorical information input to a generator.

In Tab. 5 we use different basic discriminator and training-protocol frameworks and we report results with and without our conditional batch normalization approach in the corresponding generator. Similarly to the experiments in Sec. 5.1, in our *cWC* and $cWC_{sa}$ based GANs we follow the implementation details of the reference frameworks and use their hyper-parameters (e.g., the learning rate policy, etc.) which are *not* tuned for our method. The only difference with respect to the approaches we compare with (apart from our conditional batch normalization) is a different ratio between the number of filters in the discriminator and those used in the generator in the SN + Proj. Discr. experiments (but keeping constant the total number of filters, see Sec. C.2 for more details). Tab. 5 shows that both *cWC* and $cWC_{sa}$ improve the IS and the FID values with respect to the basic frameworks in both the CIFAR-10 and the CIFAR-100 dataset. On CIFAR-10, *cWC* outperforms $cWC_{sa}$, probably because, with a small number of classes ($n$), learning class-specific coloring filters ($\Gamma_y$) is relatively easy and more informative than using a class-independent dictionary. However, with $n = 200$, as in the case of the Tiny ImageNet dataset, $cWC_{sa}$ obtains significantly better results than *cWC* (see Tab. 4). In Tab. 4, *cBN* corresponds to our re-implementation of (Miyato & Koyama (2018)), with a ResNet generator + cBN (Dumoulin et al. (2016b)) and an *SN + ResNet + Projection Discriminator* (Miyato & Koyama (2018)). Note that both *cWC* and $cWC_{sa}$ significantly improve both the IS and the FID values with respect to *cBN* on this large and challenging dataset.

In Tab. 2 (right) we compare *cWC* and $cWC_{sa}$ with other methods on CIFAR-10, where *cWC* achieves the best so far published conditioned results on this dataset. Note that *cWC* SN + Proj. Discr. in Tab. 2 (right) and in Tab. 5 indicate exactly the same framework, the only difference being the learning rate policy (see Sec. C.2). In fact, to emphasize the comparison, in Tab. 5 we adopted the same learning rate policy used by Miyato & Koyama (2018), which has been customized to our approach only in Tab. 2 (right).

Finally, similarly to the experiment shown in Fig. 2, we show in Tab. 6 that $cWC_{sa}$ leads to a much faster training with respect to *cBN*. Specifically, we use the ImageNet dataset (Deng et al. (2009)), which corresponds to a very large-scale generation task and we compare with a similar experiment performed by Miyato & Koyama (2018). The first row of Tab. 6 shows the original results reported in (Miyato & Koyama (2018)) for different training iterations. In the second row we show our results. Note that, in order to fit the generator network used by Miyato & Koyama (2018) on a single GPU, we had to reduce its capacity, decreasing the number of $3 \times 3$ convolutional filters. Hence, our generator, *including our additional coloring filters*, has 6M parameters while the generator of (Miyato & Koyama (2018)) has 45M parameters. The discriminator is the same for both methods: 39M parameters. These results show that, using a generator with only $\sim 13\%$ of the parameters of the generator used in (Miyato & Koyama (2018)), $cWC_{sa}$ can learn much faster and reach higher IS results on a big dataset like ImageNet. The importance of this experiment lies also in the fact that it emphasizes that the advantage of our method does not depend on the increase of the parameters due to the coloring layers.

Table 5: CIFAR-10/100: comparing *cWC* and $cWC_{sa}$ with different basic frameworks.

| Method | CIFAR-10 | | | CIFAR-100 | | |
|---|---|---|---|---|---|---|
| | IS | FID 5k | FID 10k | IS | FID 5k | FID 10K |
| GP + ACGAN (Gulrajani et al. (2017)) | $8.42 \pm .10$ | - | - | - | - | - |
| *cWC* GP + ACGAN (ours) | $8.80 \pm 0.13$ | - | 17.2 | - | - | - |
| $cWC_{sa}$ GP + ACGAN (ours) | $8.69 \pm 0.13$ | - | 16.8 | - | - | - |
| SN + Proj. Discr. (Miyato & Koyama (2018)) | 8.62 | 17.5 | - | 9.04 | 23.2 | - |
| *cWC* SN + Proj. Discr. (ours) | $8.97 \pm .11$ | 16.0 | 13.5 | $9.52 \pm .07$ | 20.5 | 17.2 |
| $cWC_{sa}$ SN + Proj. Discr. (ours) | $8.85 \pm .10$ | 16.5 | 13.5 | $9.80 \pm .17$ | 20.6 | 17.4 |

### 5.2.1 ABLATION STUDY

We analyze all the main aspects of the conditional coloring introduced in Sec. 4 using an *SN + ResNet + Proj. Discr.* framework for all the baselines presented here. The whitening part (when

Table 6: ImageNet: IS computed using different training iterations.

| Method | 100k | 200k | 300k | 450k |
|---|---|---|---|---|
| SN + Proj. Discr. (Miyato & Koyama (2018)) | 17.5 | 21 | 23.5 | $29.7 \pm 0.61$ |
| $cWC_{sa}$ SN + Proj. Discr. (ours) | $\mathbf{21.39} \pm \mathbf{0.37}$ | $\mathbf{26.11} \pm \mathbf{0.40}$ | $\mathbf{29.20} \pm \mathbf{0.87}$ | $\mathbf{34.36} \pm \mathbf{0.67}$ |

used) is obtained using Eq. 3. We call *cWC-cls-only* a version of $CondColoring()$ with only the class-specific term in Eq. 5. This is implemented simply removing the class-agnostic branch in Fig. 1(b) and learning a specific $(\Gamma_y, \boldsymbol{\beta}_y)$ pair per class, without class-filter soft assignment. $cWC_{sa}$-*cls-only* is the same as *cWC-cls-only* but using the soft assignment in Eq. 6. Similarly, we call *cWC-diag* the baseline in which each $\Gamma_y$ in Eq. 5 is replaced with a *diagonal* $\Gamma_y$ matrix. This corresponds to learning a set of $d$ independent *scalar* parameters for each class $y$ ($\{(\gamma_{y,k}, \beta_{y,k})\}_{k=1}^d$) to represent class-specific information similarly to cBN (see Eq. 4). In *cWC-diag* we also use the class-agnostic branch. Note that a class-agnostic branch without any kind of class-specific parameters cannot represent at all class information in the generator. As a reference, in the first row of Tab. 7 we also report the results of exactly the same framework but with our cWC replaced with cBN (Dumoulin et al. (2016b)). Moreover, we call *c-std-C* a version of our method in which feature normalization is performed using *standardization* instead of whitening and then the two coloring branches described in Sec. 4 (Eq. 5) are applied to the standardized features. Finally, *c-std-C_{sa}* corresponds to the soft-assignment version of our conditional coloring (Sec. 4.1) used together with feature standardization.

We first analyze the whitening-based baselines (second part of Tab. 7). The results reported in Tab. 7 show that both versions based on a single coloring branch (*cls-only*) are significantly worse than *cWC-diag* which includes the class-agnostic branch. On the other hand, while on CIFAR-10 *cWC-diag*, *cWC* and $cWC_{sa}$ reach comparable results, on CIFAR-100 *cWC-diag* drastically underperforms the other two versions which are based on full-coloring matrices. This shows the representation capacity of our conditional coloring, which is emphasized in CIFAR-100 where the number of classes ($n$) is one order of magnitude larger than in CIFAR-10: With a large $n$, the scalar "$\gamma$" parameters are not sufficient to represent class-specific information. Also the comparison between *cWC* and $cWC_{sa}$ depends on $n$, and the advantage of using $cWC_{sa}$ instead of *cWC* is more evident on the Tiny ImageNet experiment (see Tab.4). Finally, comparing the standardization-based versions *c-std-C* and *c-std-C_{sa}* with the corresponding whitening-based ones (*cWC* and $cWC_{sa}$), the importance of full-feature decorrelation before coloring becomes clear. Note that *c-std-C* and *c-std-C_{sa}* have exactly the same number of parameters than *cWC* and $cWC_{sa}$, respectively, which confirms that the advantage of our method does not depend only on the increase of the filter number (see also the ImageNet experiment in Tab. 6).

Table 7: Ablation study of cWC.

| | CIFAR-10 | | CIFAR-100 | |
|---|---|---|---|---|
| Method | IS | FID 10k | IS | FID 10K |
| *cBN* | $7.68 \pm .13$ | 22.1 | $8.08 \pm .08$ | 26.5 |
| *c-std-C* | $7.92 \pm .11$ | 24.37 | $7.67 \pm .08$ | 46.46 |
| *c-std-C_{sa}* | $7.93 \pm .11$ | 26.75 | $8.23 \pm .10$ | 27.59 |
| *cWC-cls-only* | $8.10 \pm .09$ | 28.0 | $7.10 \pm .06$ | 75.4 |
| $cWC_{sa}$-*cls-only* | $8.20 \pm .09$ | 23.5 | $7.88 \pm .12$ | 33.4 |
| *cWC-diag* | $8.90 \pm .09$ | 14.2 | $8.82 \pm .12$ | 20.8 |
| *cWC* | $8.97 \pm .11$ | 13.5 | $9.52 \pm .07$ | 17.2 |
| $cWC_{sa}$ | $8.85 \pm .10$ | 13.5 | $9.80 \pm .17$ | 17.4 |

# 6 WHITENING AND COLORING IN A DISCRIMINATIVE SCENARIO

In this section we plug our WC into the *classification* networks used in (Huang et al. (2018)) and we compare with their DBN (Sec. 2) in a discriminative scenario. DBN basically consists of a ZCA-based feature whitening followed by a scaling-and-shifting transform (Sec. 2). Huang et al. (2018) plug their DBN in the first layer of different ResNet architectures (He et al. (2016)) with varying depth. DBN is based also on *grouping*, which consists in computing smaller (and, hence, more stable) covariance matrices by grouping together different feature dimensions. For a fair comparison, we also plug our WC in only the first layer of each network but we do not use grouping.

Tab. 8 shows the results averaged over 5 runs. Both WC and DBN achieve lower errors than BN in all the configurations. DBN is slightly better than WC, however the absolute error difference is quite marginal. Moreover, note that the maximum error difference over all the tested configurations is lower than 1%. This empirical analysis, compared with the significant boost we obtain using WC and cWC in Sec. 5, shows that full-feature whitening and coloring is more helpful in a GAN scenario than in a discriminative setting. As mentioned in Sec. 1-2, we believe that this is due to the higher instability of GAN training. In fact, feature normalization makes the loss landscape smoother (Santurkar et al. (2018); Kohler et al. (2018)), making the gradient-descent weight updates closer to Newton updates (Huang et al. (2018); LeCun et al. (2012); S. Wiesler (2011)). Since full-feature whitening performs a more principled normalization, this helps adversarial training in which instability issues are more critical than in a discriminative scenario (Odena et al. (2018)).

Table 8: CIFAR-10 and CIFAR-100 classification error (%).

|  | CIFAR-10 | | CIFAR-100 | |
| --- | --- | --- | --- | --- |
|  | ResNet-32 | ResNet-56 | ResNet-32 | ResNet-56 |
| *BN* | 7.31 | 7.21 | 31.41 | 30.68 |
| *WC* | 7.00 | 6.60 | 31.25 | 30.40 |
| *DBN* | **6.94** | **6.49** | - | - |

## 7  CONCLUSION

In this paper we proposed a whitening and coloring transform which extends both BN and cBN for GAN/cGAN training, respectively. In the first case, we generalize BN introducing full-feature whitening and a multivariate feature re-projection. In the second case, we exploit our learnable feature re-projection after normalization to represent significant categorical conditional information in the generator. Our empirical results show that the proposed approach can speed-up the training process and improve the image generation quality of different basic frameworks on different datasets in both the conditional and the unconditional scenario.

ACKNOWLEDGMENTS

We thank the NVIDIA Corporation for the donation of the GPUs used in this project. This project has received funding from the European Research Council (ERC) under the European Unions Horizon 2020 research and innovation programme (Grant agreement No. 788793-BACKUP).

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

# A  COMPUTING THE WHITENING MATRIX USING THE CHOLESKY DECOMPOSITION

In order to compute the whitening matrix $W_B$ in Eq. 3, we first need to compute the batch-dependent covariance matrix $\Sigma_B$. Since this computation may be unstable, we use the Shrinkage estimator (Schfer & Strimmer (2005)) which is based on blending the empirical covariance matrix $\hat{\Sigma}_B$ with a regularizing matrix (in our case we use $I$, the identity matrix):

$$\Sigma_B = (1 - \epsilon)\hat{\Sigma}_B + \epsilon I, \text{ where: } \hat{\Sigma}_B = \frac{1}{m-1}\sum_{i=1}^{m}(\mathbf{x}_i - \boldsymbol{\mu}_B)(\mathbf{x}_i - \boldsymbol{\mu}_B)^\top. \tag{7}$$

Once $\Sigma_B$ is computed, we use the Cholesky decomposition (Dereniowski & Marek (2004)) in order to compute $W_B$ in Eq. 3 such that $W_B^\top W_B = \Sigma_B^{-1}$. Note that the Cholesky decomposition is unique given $\Sigma_B$ and that it is differentiable, thus we can back-propagate the gradients through our whitening transform (more details in Sec. B). Many modern platforms for deep-network developing such as TensorFlow$^{TM}$ include tools for computing the Cholesky decomposition. Moreover, its computational cost is smaller than other alternatives such as the ZCA whitening (Kessy et al. (2017)) used, for instance, in (Huang et al. (2018)). We empirically showed in Sec. 5.1.1 that, when used to train a GAN generator, our whitening transform is much faster and achieves better results than the ZCA whitening.

In more details, $W_B$ is computed using a lower triangular matrix $L$ and the following steps:

1. We start with $LL^\top = \Sigma_B$, which is equivalent to $L^{-\top}L^{-1} = \Sigma_B^{-1}$.
2. We use the Cholesky decomposition in order to compute $L$ and $L^\top$ from $\Sigma_B$.
3. We invert $L$ and we eventually get $W_B = L^{-1}$ which is used in Eq. 3.

At inference time, once the network has been trained, $W_B$ and $\boldsymbol{\mu}_B$ in Eq. 3 are replaced with the expected $W_E$ and $\boldsymbol{\mu}_E$, computed over the whole training set and approximated using the running average as in (Ioffe & Szegedy (2015)). In more detail, at each mini-batch training iteration $t$, we compute an updated version of $\Sigma_E^t$ and $\boldsymbol{\mu}_E^t$ by means of:

$$\Sigma_E^t = (1 - \lambda)\Sigma_E^{t-1} + \lambda\Sigma_B; \quad \boldsymbol{\mu}_E^t = (1 - \lambda)\boldsymbol{\mu}_E^{t-1} + \lambda\boldsymbol{\mu}_B. \tag{8}$$

When training is over, we use $\Sigma_E$ and the steps described above to compute $W_E$ (specifically, we replace $\hat{\Sigma}_B$ with $\Sigma_E$ in Eq. 7). Note that this operation needs to be done only once, after that $W_E$ and $\boldsymbol{\mu}_E$ are fixed and can be used for inference.

## A.1  COMPUTATIONAL OVERHEAD

The computational cost of $Whitening()$ is given by the sum of computing $\hat{\Sigma}_B$ ($O(md^2)$) plus the Cholesky decomposition ($O(2d^3)$), the inversion of $L$ ($O(d^3)$) and the application of Eq. 3 to the $m$ samples in $B$ ($O(md^2)$). Note that optimized matrix multiplication can largely speed-up most of the involved operations. On the other hand, $Coloring()$ is implemented as a convolutional operation which can also be largely sped-up using common GPUs. Overall, WC is $O(d^3 + md^2)$. Using a ResNet architecture with 7 WC layers, in our CIFAR-10 experiments ($32 \times 32$ images), we need 0.82 seconds per iteration on average on a single NVIDIA Titan X. This includes both the forward and the backward pass for 1 generator and 5 discriminator updates (Gulrajani et al. (2017)). In comparison, the standard BN, with the same architecture, takes on average 0.62 seconds. The ratio WC/BN is 1.32, which is a minor relative overhead. At inference time, since $W_E$ and $\boldsymbol{\mu}_E$ are pre-computed, WC is reduced to the application of Eq. 3 and the convolutional operations implementing coloring.

Concerning the conditional case, the computational cost of both versions ($cWC$ and $cWC_{sa}$) is basically the same as WC. In $cWC_{sa}$, the only additional overhead of the class-specific branch is given by the computation of $A_y^\top D$, which is $O(sd^2)$. We again point out that optimized matrix multiplication can largely speed-up most of the involved operations.

## B  BACKPROPAGATION

In our implementation, the gradient backpropagation through our WC/cWC layers is obtained by relying on the TensorFlow$^{TM}$ automatic differentiation. However, for completeness and to make our method fully-reproducible, in this section we provide a *reverse mode automatic differentiation* (AD) (Giles (2008); Ioffe & Szegedy (2015); Walter (2012)) of the $Whitening$ back-propagation steps. Note that, since (conditional and unconditional) $Coloring$ is implemented using convolution operations, back-propagation for $Coloring$ is straightforward.

In the rest of this section we adopt the notation used by Giles (2008) that we shortly present below. Given a scalar function of a matrix $f(A)$, $\bar{A}$ indicates the reverse mode AD sensitivities with respect to $A$, i.e.: $\bar{A} = \frac{\partial f(A)}{\partial A}$.

Given a batch $B$, we subtract the mean $\mu_B$ from each sample $\boldsymbol{x}_i \in B$. Reverse mode AD mean-subtraction is trivial and described in (Ioffe & Szegedy (2015)). We then stack all the mean-subtracted samples in a matrix $X \in \mathbb{R}^{d \times m}$ (where we drop the subscript $B$ for simplicity). Using this notation, the empirical covariance matrix in Eq. 7 is given by: $\Sigma = \frac{1}{m-1} X X^T$.

In the Cholesky decomposition we have: $\Sigma = L^T L$, $W = L^{-1}$. We can rewrite Eq. 3 as a matrix multiplication and obtain $Y = WX$, where $i$-th column of $Y$ is $\hat{\boldsymbol{x}}_i$.

In reverse mode AD, $\bar{Y}$ is given and we need to compute $\bar{X}$. Using the formulas in (Giles (2008)) we obtain:

$$\bar{W} = \bar{Y} X^T, \tag{9}$$

$$\bar{L} = -W^T \bar{W} W^T \tag{10}$$

Now, using the formulas in (Walter (2012)):

$$\bar{\Sigma} = \frac{1}{2} L^{-T} (P \circ L^T \bar{L} + (P \circ L^T \bar{L})^T) L^{-1} = -\frac{1}{2} W^T (P \circ \bar{W} W^T + (P \circ \bar{W} W^T)^T) W \tag{11}$$

where:

$$P = \begin{pmatrix} \frac{1}{2} & 0 & \cdots & 0 \\ 1 & \frac{1}{2} & \ddots & 0 \\ 1 & \ddots & \ddots & 0 \\ 1 & \cdots & 1 & \frac{1}{2} \end{pmatrix}$$

and $\circ$ is Hadamard product.

Referring again to (Giles (2008)), and combining the reverse sensitives with respect to $X$ from the 2 streams we get:

$$\bar{X} = \frac{2}{m-1} \bar{\Sigma} X + W^T \bar{Y}. \tag{12}$$

## C  IMPLEMENTATION DETAILS

In our experiments we use 2 basic network architectures: ResNet (Gulrajani et al. (2017)) and DCGAN (Miyato et al. (2018)). We strictly follow the training protocol details reported in (Gulrajani et al. (2017); Miyato et al. (2018)) and provided in the two subsections below. Moreover, we adopt the same architectural details suggested in (Gulrajani et al. (2017); Miyato et al. (2018)) (e.g., the number of blocks, the number of $3 \times 3$ convolutional filters, etc.), except the use of our WC/cWC layers in our generators. We use the same hyper-parameter setting of the works we compare with,

a choice which likely favours the baselines used for comparison. The only difference with respect to the architectural setup in (Gulrajani et al. (2017); Miyato et al. (2018)) is emphasized in Sec. C.2 and concerns a different ratio between the number of $3 \times 3$ convolutional filters in the generator versus the number of discriminator's convolutional filters in the SN + Proj. Discr experiments.

## C.1 UNCONDITIONAL IMAGE GENERATION EXPERIMENTS

In all our $GP$-based experiments we use Adam with learning rate $\alpha = 2e\text{-}4$, first momentum $\beta_1 = 0$ and second momentum $\beta_2 = 0.9$. The learning rate is linearly decreased till it reaches 0. In all the $GP$-based experiments we use the WGAN loss + Gradient Penalty (Gulrajani et al. (2017)), while in the $SN$-based experiment we use the hinge loss (Miyato et al. (2018)). In all the $GP$ experiments we train the networks for 100k iterations following (Gulrajani et al. (2017)). In the CIFAR-10 experiments, SN + ResNet was trained for 50k iterations (Miyato et al. (2018)). In the STL-10 experiments, we train the networks for 100k iterations following (Miyato et al. (2018)). Each iteration is composed of 1 generator update with batch size $128$ and 5 discriminator updates with batch size $64 + 64$ (i.e., $64$ generated and $64$ real images), same numbers are used in (Gulrajani et al. (2017); Miyato et al. (2018)). When training the DCGAN architectures we use two times more iterations. However, in DCGAN each iteration is composed of 1 generator update with batch size $64$ and 1 discriminator update with batch size $64 + 64$ (Miyato et al. (2018)). In all the experiments we use the original image resolution, except for STL-10 where we downsample all images to $48 \times 48$ following (Miyato & Koyama (2018)).

We provide below the architectural details of our networks. We denote with $C_d^S$ a block consisting of the following layer sequence: (1) either a WC or a cWC layer (depending on the unconditional/conditional scenario, respectively), (2) a ReLU and (3) a convolutional layer with $d$ output filters. The output of this block has the *same* (S) spatial resolution of the input. $C_d^D$ is similar to $C_d^S$, but the output feature map is *downsampled* (D): the convolutional layer uses $stride = 2$. $C_d^U$ is an *upconvolution* (U) block with fractional $stride = 0.5$. We use a similar notation for the ResNet blocks: $R_d^S$ is a ResNet block (see Fig.1(a)) which preserves spatial resolution. $R_d^D$ downsamples the feature maps and $R_d^U$ upsamples the feature maps. Also we denote with: (1) $D_k$ a dense layer with $k$ outputs, (2) $G$ a global-sum-pooling layer, (3) $F$ a flattening layer, (4) $P_h$ a layer that permutes the dimensions of the input feature map to the target spatial resolution $h \times h$. Given the above definitions, we show in Tab. 9 the architectures used for the unconditional image generation experiments.

We again emphasize that the methods we compare with in Tab. 1 have the same network structure except the use of WC layers in the generator.

Table 9: The architectures used in the unconditional image generation experiments. All the generators have a *tanh* nonlinearity at the end. We use WC before each convolutional layer of the generators.

| Method | Generator | Discriminator |
|---|---|---|
| CIFAR-10 | | |
| WC GP + ResNet | $D_{2048} - P_4 - R_{128}^U - R_{128}^U - R_{128}^U - C_3^S$ | $R_{128}^D - R_{128}^D - R_{128}^S - R_{128}^S - G - D_1$ |
| WC SN + DCGAN | $D_{8192} - P_4 - C_{512}^U - C_{256}^U - C_{128}^U - C_3^S$ | $C_{64}^S - C_{128}^D - C_{128}^S - C_{256}^D - C_{256}^S - C_{512}^D - C_{512}^S - F - D_1$ |
| WC SN + ResNet | $D_{4096} - P_4 - R_{256}^U - R_{256}^U - R_{256}^U - C_3^S$ | $R_{128}^D - R_{128}^D - R_{128}^S - R_{128}^S - G - D_1$ |
| STL-10 | | |
| WC SN + DCGAN | $D_{18432} - P_6 - C_{512}^U - C_{256}^U - C_{128}^U - C_3^S$ | $C_{64}^S - C_{128}^D - C_{128}^S - C_{256}^D - C_{256}^S - C_{512}^D - C_{512}^S - F - D_1$ |
| WC SN + ResNet | $D_{9216} - P_6 - R_{256}^U - R_{256}^U - R_{256}^U - C_3^S$ | $R_{128}^D - R_{128}^D - R_{128}^S - R_{128}^S - G - D_1$ |

## C.2 CONDITIONAL IMAGE GENERATION EXPERIMENTS

In the conditional image generation experiments, most of the hyper-parameter values are the same as in Sec. C.1. In the CIFAR-10 experiments with $GP + ACGAN$, we train our networks for 100k iterations following (Gulrajani et al. (2017)). In the CIFAR-10 and CIFAR-100 experiments with $SN$ and *Projection Discriminator* we train our networks for 50k iterations following (Miyato & Koyama (2018)). In the Tiny ImageNet experiments, we train our networks for 100k iterations.

Table 10: The architectures used in the conditional image generation experiments. All the generators have a *tanh* nonlinearity at the end. We use cWC before each convolutional layer of the generators, except for the last $C_3^S$ block in which we use WC.

| Method | Generator | Discriminator |
|---|---|---|
| | CIFAR-10 and CIFAR-100 | |
| cWC GP + ACGAN | $D_{2048} - P_4 - R_{128}^U - R_{128}^U - R_{128}^U - C_3^S$ | $R_{128}^D - R_{128}^D - R_{128}^S - R_{128}^S - G - D_1$ |
| cWC SN + Proj. Discr. | $D_{2048} - P_4 - R_{128}^U - R_{128}^U - R_{128}^U - C_3^S$ | $R_{256}^D - R_{256}^D - R_{256}^S - R_{256}^S - G - D_1$ |
| | Tiny ImageNet | |
| cWC SN + Proj. Discr. | $D_{2048} - P_4 - R_{128}^U - R_{128}^U - R_{128}^U - R_{128}^U - C_3^S$ | $R_{64}^D - R_{128}^D - R_{256}^D - R_{512}^S - R_{1024}^S - G - D_1$ |
| | ImageNet | |
| cWC SN + Proj. Discr. | $D_{2048} - P_4 - R_{128}^U - R_{128}^U - R_{64}^U - R_{32}^U - C_3^S$ | $R_{64}^D - R_{128}^D - R_{256}^D - R_{512}^D - R_{1024}^D - R_{1024}^S - G - D_1$ |

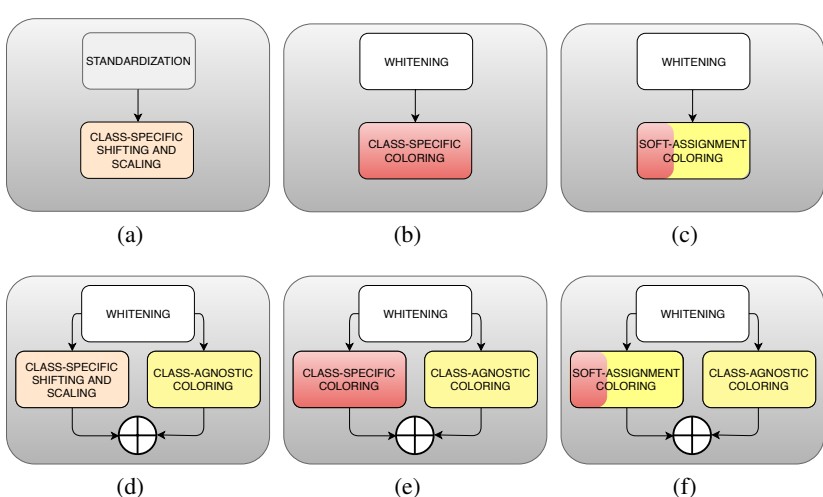

Figure 3: A schematic representation of the main baselines used in Tab. 7. (a) *cBN* (Dumoulin et al. (2016b)); (b) *cWC-cls-only*: coloring with only the conditional branch; (c) $cWC_{sa}$-*cls-only*: coloring with only the conditional branch based on the class-filter soft assignment; (d) *cWC-diag*: 2 branches and a diagonal matrix $\Gamma_y$; (e) *cWC*: plain version; (f) $cWC_{sa}$: version with soft assignment.

In the ImageNet experiment, we train our networks for 450k iterations with a starting learning rate equal to $\alpha = 5e$-4. All the other hyper-parameters are the same as in Sec. C.1. Tab. 10 shows the architectures used in the conditional image generation experiments.

Both *cWC SN + Proj. Discr.* and $cWC_{sa}$ *SN + Proj. Discr.* used in CIFAR-10 and CIFAR-100 (see Tab. 5) have an opposite generator/discriminator filter-number ratio with respect to the ratio used in the original *SN + Proj. Discr.* by Miyato & Koyama (2018) ($128/256$ vs. $256/128$). In fact, we decrease the number of $3 \times 3$ convolutional filters in each generator block from 256, used in (Miyato & Koyama (2018)), to 128. Simultaneously, we increase the number of $3 \times 3$ convolutional filters in the discriminator from 128 (Miyato & Koyama (2018)) to 256. *This is the only difference between our cWC-based frameworks and (Miyato & Koyama (2018)) in the experiments of Tab. 5.* The reason behind this architectural choice is that we hypothesize that our cWC generators are more powerful than the corresponding discriminators. Using the same generator/discriminator filter ratio as in (Miyato & Koyama (2018)), we perform on par with *SN + Proj. Discr.* (Miyato & Koyama (2018)). In Tab. 7, *cBN* refers to *SN + Proj. Discr.* with the same aforementioned $128/256$ filter ratio and used in the cWC baselines.

In Fig. 3 we show a schematic representation of the main baselines reported in Tab. 7.

Finally, in Tab. 2 (right) and in Tab. 5, *cWC SN + Proj. Discr.* indicates exactly the same framework, the only difference being a different learning rate policy. The results of *cWC SN + Proj. Discr.* reported in Tab. 5 have been obtained using the same learning rate policy used by Miyato & Koyama (2018). However, our model already achieved its peak performance after 30k iterations. Hence, we

decreased the learning rate by a factor of 10 at iteration 30k and then we trained *cWC SN + Proj. Discr.* for additional 1k iterations. This brought to a performance boost, as reported in Tab. 2 (right).

## C.3 EVALUATION METRICS

In our experiments we use 2 evaluation metrics: Inception Score (IS) (Salimans et al. (2016)) and the Fréchet Inception Distance (FID) (Heusel et al. (2017)). These metrics are the two most widely adopted evaluation criteria for generative methods. While most of generative works report quantitative results using only one of these two criteria, we chose to report them both to show the agreement of these two metrics: in most of the presented comparisons and ablation studies, when IS is larger, FID is lower.

IS was originally introduced by Salimans et al. (2016), who suggest to measure IS using 5k generated images, repeat this generation 10 times and then average the results. We follow exactly the same protocol.

FID is a more recent metric and the evaluation protocol is less consolidated. Miyato & Koyama (2018) suggest to compute FID using 10k images from the test set together with 5k generated images. Heusel et al. (2017) suggest to use a sample sizes of at least 10k real and 10k generated images. In all our tables, FID 5k refers to the former protocol (Miyato & Koyama (2018)) and FID 10k refers to the second protocol (Heusel et al. (2017)).

## D    USER STUDY

In order to further emphasize the improvement we have when using $cWC_{sa}$ instead of cBN, we show in Tab. 11 a user study performed with 21 persons. Specifically, the dataset used is Tiny ImageNet and the cBN baseline is the same used for the experiments reported in Tab. 4. We show to each user 50 paired image-sets (using 50 randomly selected classes), generated using either $cWC_{sa}$ or cBN. Each image-set consists of 16 stacked images of the same class, used to increase the robustness of the user's choice. The user selects the image-set which he/she believes is more realistic. These results clearly show the superiority of $cWC_{sa}$ against cBN.

Table 11: Tiny ImageNet user-study. Comparing cBN and $cWC_{sa}$ using human judgments.

|  | User-averaged class-based preference for $cWC_{sa}$ | Cronbach's alpha inter-user agreement |
|---|---|---|
| *cBN vs cWC$_{sa}$* | 67.34 % | 0.87 (Good) |

## E    COMPARISON BETWEEN THE CHOLESKY AND THE ZCA-BASED WHITENING PROCEDURES

In this section we compare the stability properties of our Cholesky decomposition with the ZCA-based whitening proposed in (Huang et al. (2018)). Specifically, we use $W_{zca}C$ which is the variant of our method introduced in Sec. 5.1.1, where we replace our Cholesky decomposition with the ZCA-based whitening. In Fig. 4 we plot the IS/training-iteration curves corresponding to both $WC$ and $W_{zca}C$. This graph shows that $W_{zca}C$ is highly unstable, since training can frequently degenerate. When this happens, $W_{zca}C$ collapses to a model that always produces a constant, uniform grey image.

The reason for this behaviour is likely related to the way in which the gradients are backpropagated through the ZCA-based whitening step. Specifically, following (Huang et al. (2018)), the gradient of the loss with respect to the covariance matrix depends from the inverse of the difference of the covariance-matrix singular values (see (Huang et al. (2018)), Appendix A.2). As a consequence, if some of the singular values are identical or very close to each other, this computation is ill-conditioned. Conversely, our whitening procedure based on the Cholesky decomposition (Sec. A) has never showed these drastic instability phenomena (Fig. 4).

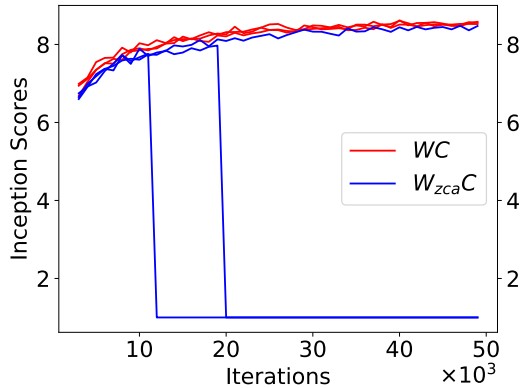

Figure 4: CIFAR-10: stability comparison between $WC$ and $W_{zca}C$. While $WC$ is stable, in our experiments we observed that $W_{zca}C$ can frequently degenerate to a model which produces a constant, gray image. When this happens, the IS value collapses. Specifically, this figure shows three randomly initialized training runs for both $WC$ and $W_{zca}C$. In two out of three runs, $W_{zca}C$ degenerated at some point during training.

## F  EXPERIMENTS USING SMALL-SIZE DATASETS

In this section we show additional experiments using two small-size datasets: the MNIST (LeCun et al. (1998)) and the Fashion-MNIST dataset (Xiao et al. (2017)) (see Fig. 9 for some generated image examples). In both cases we use the *SN + ResNet* architecture described in Tab. 12. On MNIST we train for 5k iterations with a constant learning rate, while on Fashion-MNIST we train for 20k iterations with a linearly decaying learning rate (see Sec. C.1). All the other hyper-parameters are the same as in Sec. C.1.

The results are reported in Tab. 13. Unconditional image generation on MNIST is the only setup in which we have not observed an improvement using our WC. However, on the more challenging Fashion-MNIST dataset, we did observe an improvement, especially in the conditional scenario.

Table 12: The MNIST and the Fashion-MNIST architectures. Similarly to the architectures in Tab 9-10: (1) All the generators have a $tanh$ nonlinearity at the end, (2) We use WC/cWC before each convolutional layer of the generators, (3) In the conditional case, in the last $C_3^S$ block, cWC is replaced by WC.

| Method | Generator | Discriminator |
|--------|-----------|---------------|
| WC | $D_{12544} - P_7 - R_{256}^U - R_{256}^U - C_3^S$ | $R_{128}^D - R_{128}^D - R_{128}^S - R_{128}^S - G - D_1$ |
| cWC | $D_{6272} - P_7 - R_{128}^U - R_{128}^U - C_3^S$ | $R_{128}^D - R_{128}^D - R_{128}^S - R_{128}^S - G - D_1$ |

Table 13: FID 10k scores for the unconditional and the conditional image generation experiments on the MNIST and the Fashion-MNIST datasets.

| Unconditional | | | Conditional | | |
|--------|-------|---------------|--------|-------|---------------|
| Method | MNIST | Fashion-MNIST | Method | MNIST | Fashion-MNIST |
| BN | 3.6 | 10.6 | cBN | 4.5 | 7.5 |
| WC | 4.9 | 10.4 | cWC | 4.3 | 6.2 |

## G  QUALITATIVE RESULTS

In this section we show some qualitative results obtained using our WC/cWC methods. Specifically, examples of images generated using the CIFAR-10 dataset are shown in Fig. 5. Fig. 6 (a)-6 (b)

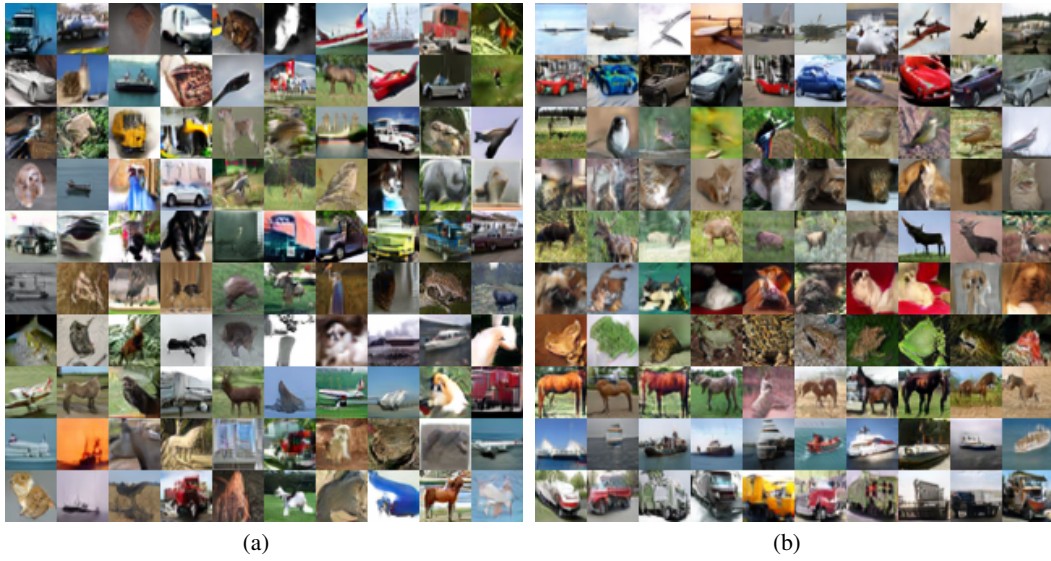

(a)                                                   (b)

Figure 5: CIFAR-10 images generated using: (a) *WC SN + ResNet* and (b) *cWC SN + ResNet* (samples of the same class displayed in the same row).

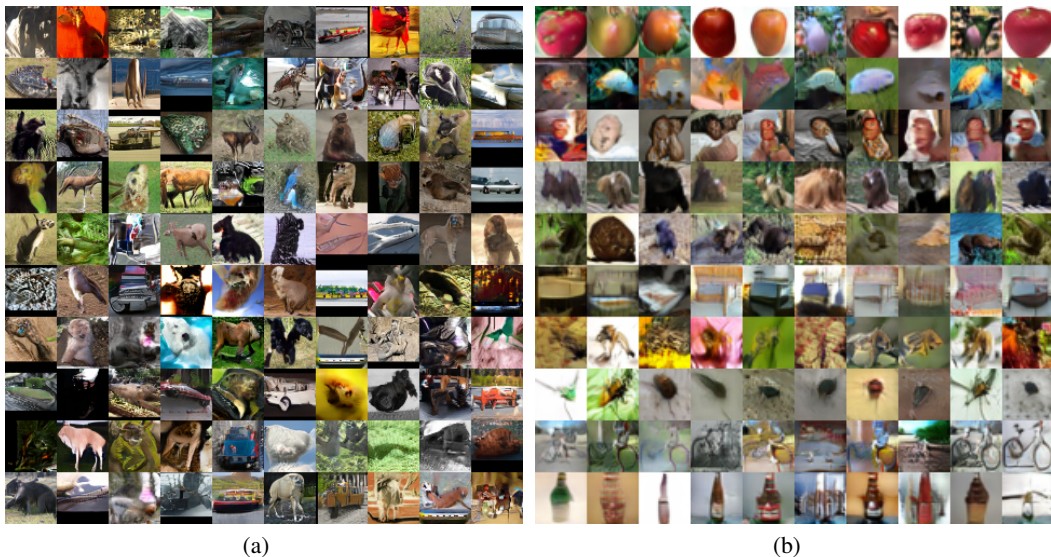

(a)                                                   (b)

Figure 6: Qualitative results: (a) STL-10 images generated using WC and (b) CIFAR-100 images generated using $cWC_{sa}$ (first 10 CIFAR-100 classes, samples of the same class are displayed in the same row).

show examples generated on the STL-10 and the CIFAR-100 dataset, respectively. In Fig. 8(a)-8(b) we show Tiny ImageNet and ImageNet samples, respectively. Fig. 9 shows MNIST and Fashion-MNIST generated images.

Finally, in Fig. 7 we show some images generated using CIFAR-10 and CIFAR-100 in which the value of the noise vector $\mathbf{z}$ is kept fixed while varying the conditioning class $y$.

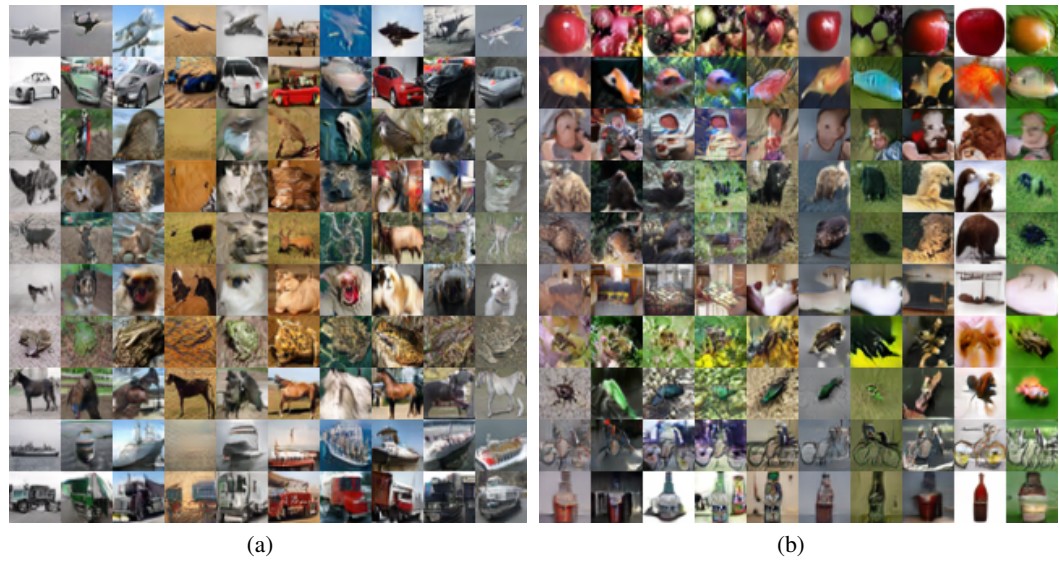

(a)                                                          (b)

Figure 7: Images generated using *cWC SN + ResNet*. In each column the value of the noise vector **z** is kept fixed, while changing in each row the class-label $y$. (a) CIFAR-10 and (b) CIFAR-100.

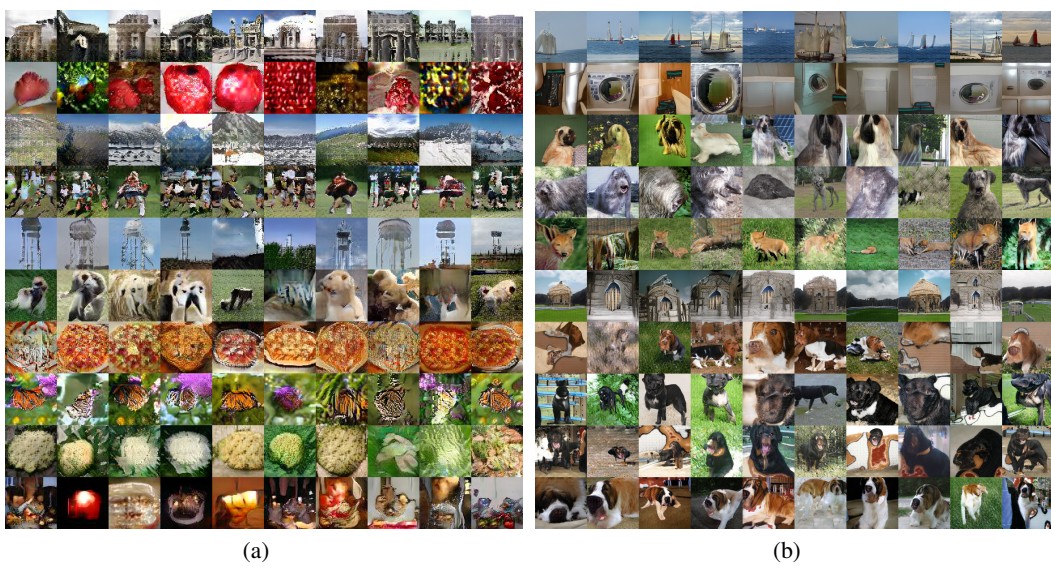

(a)                                                          (b)

Figure 8: Tiny ImageNet (a) and ImageNet (b) images generated using $cWC_{sa}$ by randomly selecting 10 classes (samples of the same class are displayed in the same row).

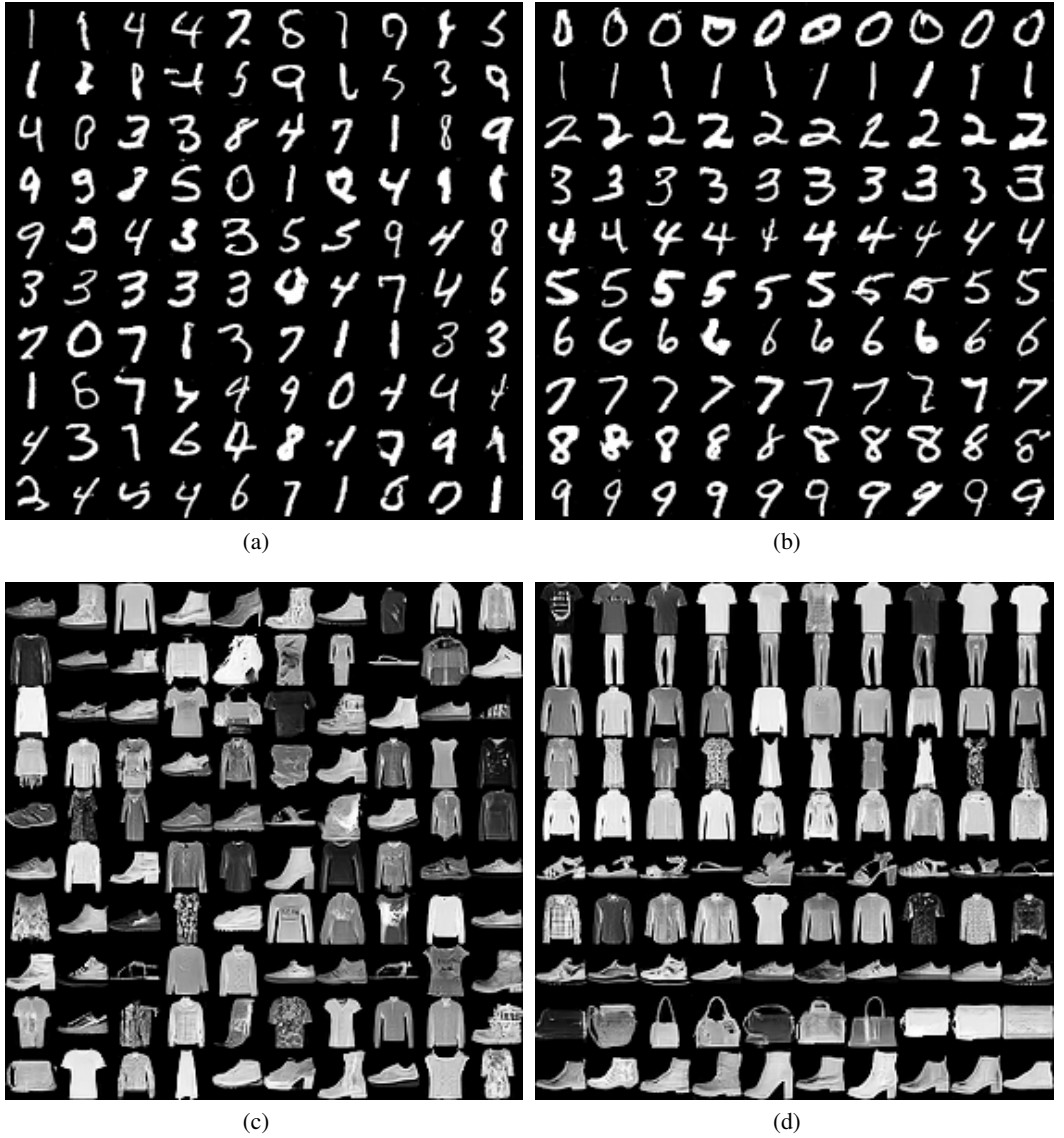

Figure 9: MNIST (a, b) and Fashion-MNIST (c, d) images generated using WC (a, c) and cWC (b, d).

