# OpenReview forum: "Whitening and Coloring Batch Transform for GANs"
_ICLR.cc/2019/Conference_

### Official Review · AnonReviewer2 · 2018-10-28
**Good results, motivation unclear for GAN**

**Rating:** 7
**Confidence:** 4

**Review:**

This paper proposed Whitening and Coloring (WC) transform to replace batch normalization (BN) in generators for GAN. WC generalize BN by normalizing features with decorrelating (whitening) matrix, and then denormalizing (coloring) features by learnable weights. The main advantage of WC is that it exploits the full correlation matrix of features, while BN only considers the diagonal. WC is differentiable and is only 1.32x slower than BN. The authors also apply conditional WC, which learn the parameters of coloring conditioned on labels, to conditional image generation.  Experimental results show WC achieves better inception score and FI distance comparing to BN on CIFAR-10, CIFAR-100, STL-10 and Tiny Imagenet. Furthermore, the conditional image generation results by WC are better than all previous methods.

I have some detailed comments below.

+ The paper is well written, and I generally enjoyed reading the paper.
+ The experimental results look sufficient, and I appreciate the ablation study sections.
+ The score on supervised CIFAR-10 is better than previous methods.

- The main text is longer than expectation. I would suggest shorten section 3.1 Cholesky decomposition, section 4 conditional color transformation and the text in section 5 experiments.
- The proposed WC transform is general. It is a bit unclear why it is particularly effective for generator in GAN. Exploiting the full correlation matrix sounds reasonable, but it may also introduce unstability. It would help if the authors have an intuitive way to show that whitening is better than normalization.
- It is unclear why conditional WC can be used for generation conditioned on class labels. In Dumoulin 2016, conditional instance normalization is used for generating images conditioned on styles. As image styles are described by Gram matrix (correlation) of features, changing first order and second order statistics of features is reasonable for image generation conditioned on styles. I cannot understand why conditional WC can be used for generation conditioned on class labels. I would like the authors to carefully explain the motivation, and also provide visual results like using the same random noise as input, but only changing the class conditions.
- It is unclear to me why the proposed whitening based on Cholesky decomposition is better than ZCA-based in Huang 2018. Specifically, could the authors explain why WC is better than W_{aca}C in Table 3?
- The authors claim progressive GAN used a larger generator to achieve a better performance than WC. The WC layer is generally larger than BN layer and has more learnable parameters. Could the authors compare  the number of parameter of generator in BN-ResNet, WC-ResNet, and progressive GAN?
- In Table 3, std-C is better than WC-diag, which indicates coloring is more important. In Table 6, cWC-diag is better than c-std-C, which indicates whitening is more important. Why?
- What is the batch size used for training? For conditional WC, do the samples in each minibatch have same label?
- Having ImageNet results will be a big support for the paper.


===========  comments after reading rebuttal ===========

I appreciate the authors' feedback. I raised my score for Fig 7 showing the conditional images, and for experiments on ImageNet.

I think WC is a reasonable extension to BN, and I generally like the extensive experiments. However, the paper is still borderline to me for the following concerns.

- I strongly encourage the authors to shorten the paper to the recommended 8-page.

- The motivation of WC for GAN is still unclear. WC is general extension of BN, and a simplified version has been shown to be effective for discrimination in Huang 2018. I understand the empirically good performance for GAN. But I am not convinced why WC is particularly effective for GAN, comparing to discrimination. The smoothness explanation of BN applies to both GAN and discrimination. I actually think it may be nontrivial to extend the smoothness argument from BN to WC.

- The motivation of cWC is still unclear. I did not find the details of cBN for class-label conditions, and how they motivated it in (Gulrajani et al. (2017) and (Miyato et al. 2018). Even if it has been used before, I would encourage the authors to restate the motivation in the paper. Saying it has been used before is an unsatisfactory answer for an unintuitive setting.

- Another less important comment is that it is still hard to say how much benefits we get from the more learnable parameters in WC than BN. It is probably not so important because it can be a good trade-off for state-of-the-art results. In table 3 for unconditioned generation, it looks like the benefits come a lot from the larger parameter space. For conditioned generation in table 6, I am not sure if whitening is conditioned or not, which makes it less reliable to me. If whitening is conditioned, then the samples in each minibatches may not be enough to get a stable whitening. If whitening is unconditioned, then there seems to be a mismatch between whitening and coloring.

====== second round after rebuttal =============
I raise the score again for the commitment of shortening the paper and the detailed response from the authors. That being said, I am not fully convinced about motivations for WC and cWC.

- GAN training is more difficult and unstable, but that does not explain why WC is particularly effective for GAN training.

- I have never seen papers saying cBN/cWC is better than other conditional generator conditioned on class labels. I think the capacity argument is interesting, but I am not sure if it applies to convolutional net (where the mean and variance of a channel is used), or how well it can explain the performance because neural nets are overparameterized in general. I would encourage authors to include these discussions in the paper.

---

> ### Author Response · Authors · 2018-11-26
> **Response to Reviewer #2 (Part 1)**
>
> Thank you for your detailed review. Below our answers.
>
> Q: The main text is longer than expectation. I would suggest shorten section 3.1 Cholesky decomposition, section 4 conditional color transformation and the text in section 5 experiments.
>
> A: Thank you for your suggestion. It is not clear to us how much the new version can be different from the submitted one, however, in the final version we will shorten the paper and possibly move Sec. 3.1 and 3.2 to the Appendix, if you think they are less important.
>
> -----
>
> Q: The proposed WC transform is general. It is a bit unclear why it is particularly effective for generator in GAN. Exploiting the full correlation matrix sounds reasonable, but it may also introduce unstability. It would help if the authors have an intuitive way to show that whitening is better than normalization.
>
> A: Please, see our answer to Reviewer #1 concerning the motivation behind the choice of a GAN scenario and our first answer to Reviewer #3 concerning a more intuitive explanation of why whitening is better than standardization.
>
> -----
>
> Q: It is unclear why conditional WC can be used for generation conditioned on class labels. In Dumoulin 2016, conditional instance normalization is used for generating images conditioned on styles. As image styles are described by Gram matrix (correlation) of features, changing first order and second order statistics of features is reasonable for image generation conditioned on styles. I cannot understand why conditional WC can be used for generation conditioned on class labels. I would like the authors to carefully explain the motivation, and also provide visual results like using the same random noise as input, but only changing the class conditions.
>
>
> A: Since the introduction of the cBN in (Dumoulin et al. (2016b)), many works showed that this is a very powerful method for representing object-class information in conditional GANs (e.g., (Gulrajani et al. (2017)); (Miyato et al. (2018))). We basically extend this idea using (class-specific) convolutional filters instead of (class-specific) scaling  parameters (Sec. 1, 2). Note that G needs information about y. Our conditional coloring filters implicitly represent y by projecting the features into class-specific distributions. In other words, all the layers of our generator share the weights across all the classes, except for the convolutional filters in the conditional-coloring layers. This is enough for the network to influence the image generation process with respect to a specific categorical variable (y).
> In Sec. 2 we briefly describe other methods used in the literature to condition the generation process on a specific class label.
> Following your suggestion, in the new version of the paper we added the new Fig. 7 in Appendix F in which we keep fixed the value of z (the noise vector) and we vary y.
>
> -----
>
> Q: It is unclear to me why the proposed whitening based on Cholesky decomposition is better than ZCA-based in Huang 2018. Specifically, could the authors explain why WC is better than W_{aca}C in Table 3?
>
> A: Please, see our second answer to Reviewer #3
>
> -----
>
> Q: The authors claim progressive GAN used a larger generator to achieve a better performance than WC. The WC layer is generally larger than BN layer and has more learnable parameters. Could the authors compare  the number of parameter of generator in BN-ResNet, WC-ResNet, and progressive GAN?
>
> A: You are right: both WC and cWC have more learnable weights, due to the coloring step. However, note that our performance boost does not depend only on the increase of the  convolutional-filter number. For instance, if you compare c-std-C and c-std-C_{sa} with the corresponding whitening-based versions cWC and cWC_{sa}, having exactly the same number of parameters, you see that the latters drastically outperform the former ones (Tab. 6). We emphasized this in the new paper at the end of Sec. 5.2.1. Please, see also below our answer about the ImageNet experiment.
>
> The number of overall parameters used in the architectures you mentioned are:
> BN-ResNet (called SN + ResNet + Proj. Discr. in the paper). G: 4.3M; D: 1M.
> WC-ResNet (called WC + SN + ResNet + Proj. Discr. in the paper). G: 4.7M; D: 1M.
> Progressive GAN. G: 18.9M; D: 18.9M.
>
> where G indicates the Generator and D the Discriminator.

---

> > ### Author Response · Authors · 2018-11-26
> > **Response to Reviewer #2 (Part 2)**
> >
> > Q:  In Table 3, std-C is better than WC-diag, which indicates coloring is more important. In Table 6, cWC-diag is better than c-std-C, which indicates whitening is more important. Why?
> >
> > A: The results in both Tab. 3 and Tab. 6 show that whitening and coloring reach the highest accuracy when combined. The two settings (conditional and unconditional) are different and they are not easy to compare to each other. However, we believe that the reason for which the relative importance of the two steps is different in these two settings is probably related to the "effective batch size", which is, for the conditional case, m/n, being "n" the number of classes (see Sec. 4). In other words, the Gamma filters (unconditional case) are trained with "m" samples at each iteration, while the Gamma_y filters are trained with only "m/n" samples". As a consequence, the Gamma_y filters are trained with less data, making the coloring step relatively less effective than in the unconditional case.
> >
> > -----
> >
> > Q: What is the batch size used for training? For conditional WC, do the samples in each minibatch have same label?
> >
> > A: The batch size used for training depends on the specific experiments since we strictly adopted the protocols and the hyper-parameters of each framework we compare with. All the details, batch size included, are in Appendix A.1. In cWC, the samples in each mini-batch do not have the same label. On average, there are m/n instances with the same label in a batch, being "n" the cardinality of the class set. This is called the "effective batch size". For details, please, see Sec. 4.
> >
> > -----
> >
> > Q: Having ImageNet results will be a big support for the paper.
> >
> > A: Following your suggestion, we are training our WC + SN + ResNet + Proj. Discr. on ImageNet. Up to now the complete training is not yet finished because the dataset is very large and the basic network (based on the SN + ResNet + Proj. Discr. approach) used in (Miyato et al. (2018)) for a similar experiment is very big (i.e., too big to be fitted in a single GPU). We will have final results at camera-ready time. However, we have already almost reached the performance of (Miyato et al. (2018)) with much less iterations and with a much smaller capacity network. Specifically:
> >
> > - At 100k iterations, ours (WC + SN + ResNet  + Proj. Discr) has an IS value of 21.4, while (Miyato et al. (2018)) is 17.5.
> > - At 200k iterations, ours is 26.12, while (Miyato et al. (2018)) is 21.
> > - At 300k iterations, ours is 29.19, while (Miyato et al. (2018)) is 23.5.
> >
> > (Miyato et al. (2018)) report results at 450k iterations. Note that with only 2/3 iterations we are already almost on par with their results:
> > - At 300k iterations, ours is 29.19, while at 450k iterations (Miyato et al. (2018)) is 29.5.
> >
> > Moreover, note that, in order to fit the basic network on a single GPU, we had to reduce its capacity, decreasing the number of 3X3 convolutional filters. All in all, and considering our additional coloring filters, our generator has 6M parameters while the generator of (Miyato et al. (2018)) has 45M parameters. The discriminator is the same for both: 39M.
> > Thus this preliminary results show that with 2/3 iterations and with a generator with overall only 13% of the parameters of the generator used in (Miyato et al. (2018)) we are almost on par on a big dataset like ImageNet. We will add this experiment in the final version of the paper, which we believe it is important also to emphasize that the advantage of our method does not depend on the increase of the parameters due to the coloring layers.

---

> ### Author Response · Authors · 2018-11-29
> **Reply to the after-rebuttal comments (first part)**
>
> Thank you for appreciating our response and our WC proposal. Below our answers to your feedback.
>
> Q: I strongly encourage the authors to shorten the paper to the recommended 8-page.
>
> A: We will move Sec. 3.1 and 3.2 to the Appendix and shorten the whole paper.
>
> -----
> Q: The motivation of WC for GAN is still unclear. WC is general extension of BN, and a simplified version has been shown to be effective for discrimination in Huang 2018. I understand the empirically good performance for GAN. But I am not convinced why WC is particularly effective for GAN, comparing to discrimination. The smoothness explanation of BN applies to both GAN and discrimination. I actually think it may be nontrivial to extend the smoothness argument from BN to WC.
>
> A: We agree that the need of loss smoothness applies to both GAN and discriminative networks. However, the reason for which in an adversarial setting stability is more important is related to fact that GAN optimization aims at finding a Nash equilibrium between two players, a problem which is more difficult and less stable than the common discriminative-network optimization and that frequently leads to non-convergence.
> We plan to provide an empirical evidence that the GAN loss is smoother using our WC than when using BN in a Journal extension of this paper following the protocol suggested by Odena et al.(2018).

---

> > ### Author Response · Authors · 2018-11-29
> > **Reply to the after-rebuttal comments (second part)**
> >
> > Q: The motivation of cWC is still unclear. I did not find the details of cBN for class-label conditions, and how they motivated it in (Gulrajani et al. (2017) and (Miyato et al. 2018). Even if it has been used before, I would encourage the authors to restate the motivation in the paper. Saying it has been used before is an unsatisfactory answer for an unintuitive setting.
> >
> > A: The intuitive idea behind class-specific parameters Gamma_y, beta_y in cWC (and gamma_y, beta_y in cBN) is that they are used to separate samples accordingly to the desired class y. In other words, G needs information to understand if a dog or a cat should be synthesized. Using (Gamma_y, beta_y), each feature x (generated from an initial noise vector z) is projected into a specific distribution. The following 3x3 convolutional layer can exploit this y-specific part of the feature space to "understand" that shape/appearance specific for the y class should be produced. We agree that an intuitive explanation is helpful for the reader and we will add this explanation to the final version of the paper.
> >
> > Note that, among the three known alternatives used to provide G information about y, i.e.: (1) concatenation of (a one-hot representation of) y with z, (2) cBN and (3) our proposed cWC, the latter is the most general.
> > Indeed, using a simplified one-linear-layer G, it can be shown that (1)-(3) above correspond to:
> >
> > (1) G_concat(z,y) ~ N (m_y, S). The distribution which can be produced by G is normal with a class-specific mean (m_y) but with a class-independent covariance matrix S.
> > (2) G_cBN(z,y) ~ N(m_y, diag(S_y) * S). G generates samples with a class-specific mean and class-specific variance values but with a class-independent correlation matrix.
> > (3) G_cWC(z,y) ~ N(m_y, S_y). G generates distributions with both means and covariance matrices which are class-specific.
> >
> > Where we assume z ~ N(0, I).
> > This shows that our cWC improves the representation capacity of a given linear layer. Similarly, each block of a deep-network generator gains representation capacity using cWC (please, see also the difference between Eq. (7) and (6) in the paper).
> >
> > -----
> >
> > Q: Another less important comment is that it is still hard to say how much benefits we get from the more learnable parameters in WC than BN. It is probably not so important because it can be a good trade-off for state-of-the-art results. In table 3 for unconditioned generation, it looks like the benefits come a lot from the larger parameter space. For conditioned generation in table 6, I am not sure if whitening is conditioned or not, which makes it less reliable to me. If whitening is conditioned, then the samples in each minibatches may not be enough to get a stable whitening. If whitening is unconditioned, then there seems to be a mismatch between whitening and coloring.
> >
> > A: The Whitening() procedure used in the conditional case (cWC) is exactly the same procedure used in the unconditional setting (WC) and it is not conditioned by y, meaning that the covariance matrix is computed using all the batch samples, independently of the class. This is done for the reason you mention: we don't want to have covariance matrices computed with only m/n samples, which may be unstable. We will clarify this in the paper. It is not completely clear to us what you mean with "mismatch". If you mean that feature separation is performed only by CondColoring(), that is true: Whitening() is identical in both WC and cWC. However we believe that this is a good trade-off, being the (Gamma_y, Beta_y) parameters in CondColoring() enough to perform feature separation (please, see also our previous answer). Note that also in cBN, the standardization part is unconditioned.

---

> > > ### Comment · AnonReviewer2 · 2018-11-29
> > > **thanks for clarification on whitening, don't understand the distribution argument for cWC**
> > >
> > > Thanks for clarifying the whitening is not conditioned. I will remove that part from review later. Again, I think it is a tolerable trade-off, and it is not the main reason for the evaluation and score.
> > >
> > >
> > > I don't quite get your explanation about distribution of G_concat(z,y), G_cBN(z,y) , G_cWC(z,y). Why are the output distributions of generator Gaussian? Even though the covariance of activations are not explicitly changed when we use concatenation as condition, why is the output covariance independent of labels?

---

> > > > ### Author Response · Authors · 2018-11-30
> > > > **More details on the distribution argument for cWC**
> > > >
> > > > Q: I don't quite get your explanation about distribution of G_concat(z,y), G_cBN(z,y) , G_cWC(z,y). Why are the output distributions of generator Gaussian? Even though the covariance of activations are not explicitly changed when we use concatenation as condition, why is the output covariance independent of labels?
> > > >
> > > > A: Note that our conclusions about the Gaussian distributions of G, etc. are drawn using some simplifying assumptions, such as the fact of  having a simplified one-linear-layer G. We provide below a formal demonstration, and then we extend the conclusions to a deep, non-linear G.
> > > >
> > > > Assumptions: G is a one-linear-layer network, specifically: G(z) = W * z + b; and z ~ N(0, I). W is a weight matrix, z and b are vectors.
> > > >
> > > > (1) G_concat(z,y) = U * [z || one_hot(y)] + b; where U = [W -- V] and:
> > > > one_hot(y) is the one-hot representation of the integer variable y,
> > > > || denotes vertical concatenation,
> > > > -- denotes horizontal concatenation,
> > > > V is an additional parameter matrix corresponding to the one_hot(y) concatenation.
> > > >
> > > > Hence:
> > > > G_concat(z,y) = W * z + V * one_hot(y) + b.
> > > >
> > > > Since we consider only linear operations and we assume that z ~ N(0, I), the output distribution will also be Gaussian. Note that the covariance matrix of the generated distribution depends only on the first term (W * z), being the latter constant with respect to z. Specifically, the covariance matrix depends on W, which is class-independent.
> > > >
> > > > (2) G_cBN(z,y) = W * (gamma_y * z + beta_y) + b, where:
> > > > gamma_y is a diagonal matrix whose (k,k)-th element corresponds to the gamma_y,k scaling parameter in Eq. (6),
> > > > beta_y is the vector of the shifting parameters
> > > >
> > > > Note that standardization is omitted because here we are modeling only the representation capacity of the scaling-shifting parameters of cBN.
> > > > The covariance matrix in this case depends on W * gamma_y. This leads to class-specific variance values but with a class-independent correlation matrix.
> > > >
> > > > (3) G_cWC(z,y) = W * (Gamma_y * z + beta_y) + b, where we omit the class-agnostic part of Eq. (7) for simplicity and:
> > > > Gamma_y is the coloring matrix defined in Sec. 4.
> > > >
> > > > Also in this case the covariance matrix depends on W * Gamma_y, but now Gamma_y is a full-matrix, which leads to a class-specific covariance matrix.
> > > > ------
> > > >
> > > > Similar arguments hold also for non-normal distributions. Moreover, it can be shown that, given a one-linear layer in a deep G,  there is a hierarchy on the representation capacity among G_concat, G_cBN and G_cWC. Hence, comparing two networks G_1 and G_2 having the same architecture (i.e., same number of layers, etc.), if all the layers in G_1 have a higher representation capacity than the corresponding layers in G_2, than, overall, G_1 has a greater or equal representation capacity than G_2.
> > > >
> > > > (Gulrajani et al. (2017)) and (Miyato et al. (2018)) empirically showed that this inequality is strict, in a sense that G_cBN(z,y) is better than G_concat(z,y). In our work we empirically show that G_cWC(z,y) is better than G_cBN(z,y) (e.g., see Tab. 6).

---

> > > > > ### Comment · AnonReviewer2 · 2018-11-30
> > > > > **Not convinced**
> > > > >
> > > > > I agree with the authors on one-layer linear network. However, I think multi-layer nonlinear network can not be simplified as linear operator. The generator is supposed to approximate 'any' distribution.
> > > > >
> > > > > Could the authors kindly point out a reference where the generator is treated as linear operator, or the output distribution is considered Gaussian?
> > > > >
> > > > > Even for one-layer nonlinear NN, the expressive power is good. Let's say we have a Gaussian variable x~N(0, 1). What is the variance for variable y = [x-\mu]_{+}, which is basically shift the mean and then use ReLU?
> > > > >
> > > > > Could the authors kindly point out to me where do (Gulrajani et al. (2017)) and (Miyato et al. (2018)) talk about G_cBN(z,y) and G_concat(z,y)?

---

> > > > > > ### Author Response · Authors · 2018-12-04
> > > > > > **Answer**
> > > > > >
> > > > > > Q: I agree with the authors on one-layer linear network. However, I think multi-layer nonlinear network can not be simplified as linear operator. The generator is supposed to approximate 'any' distribution.
> > > > > > Could the authors kindly point out a reference where the generator is treated as linear operator, or the output distribution is considered Gaussian?
> > > > > > Even for one-layer nonlinear NN, the expressive power is good. Let's say we have a Gaussian variable x~N(0, 1). What is the variance for variable y = [x-\mu]_{+}, which is basically shift the mean and then use ReLU?
> > > > > >
> > > > > >
> > > > > > A: Sorry but this question is not clear to us. Of course deep generators cannot be simplified as a single linear layer. That assumption was used only to simplify the demonstration. Indeed, the extension of the same formal proof to multiple, non-linear-layers G, is much harder.
> > > > > >
> > > > > > Note that the conclusions we draw from that demonstration can be easily extended to non-normal input distributions and for a possible intermediate (linear) layer of G.
> > > > > > The intention of our demonstration was to show that, among all the common class-conditioned input transformations performed in a single linear layer of G, concatenation is less expressive than cBN which is less expressive than cWC.
> > > > > >
> > > > > > The conclusions over a more realistic, deep G (let's call it "G_deep"), which we draw after the end of the formal proof, are based on the following intuition. If we assume that G_deep receives information about the class y using one over these 3 mechanisms (G_concat, G_cBN or G_cWC), then G_deep is composed of: (1) layers which are shared over all the classes; (2) class-specific layers (i.e., one over G_concat, G_cBN or G_cWC). The layers shared over all the classes are class-independent (by definition), so they do not bring any information about y. The class-specific layers have a representation capacity which depend on the specific chosen mechanism (G_concat, G_cBN or G_cWC). Hence the conclusions: G_deep based on G_cBN has a greater or equal representation capacity than a corresponding G_deep based on G_concat, etc.
> > > > > >
> > > > > > ------------------
> > > > > > Q: Could the authors kindly point out to me where do (Gulrajani et al. (2017)) and (Miyato et al. (2018)) talk about G_cBN(z,y) and G_concat(z,y)?
> > > > > >
> > > > > > A:  Miyato et al. (2018) explain in the beginning of Sec. 5.1 of their paper that they use the cBN proposed in (Dumoulin et al. (2016b)).
> > > > > >
> > > > > > In (Gulrajani et al. (2017)) cBN is not explicitly referred to in the paper but it is used in the publicly available code (https://github.com/igul222/improved_wgan_training/blob/master/gan_cifar_resnet.py#L79) for the supervised experiments (corresponding to Tab. 3-right of their paper).
> > > > > >
> > > > > > None of the aforementioned papers directly perform ablation studies on this aspect (being not an original contribution of their method). However, for instance, Gulrajani et al. (2017) compare their method with other GANs using class-label concatenation in Tab. 3-right of their paper (which largely corresponds to Tab. 2-right of our paper). The low-ranked methods in that table (e.g., SteinGAN (Wang & Liu (2016)), DCGAN with labels (Wang & Liu (2016)), AC-GAN (Odena et al. (2016))) correspond to concatenation-based approaches.

---

> > > > > > > ### Comment · AnonReviewer2 · 2018-12-06
> > > > > > > **one-layer linear is too simple**
> > > > > > >
> > > > > > > It is not necessarily to do the real complicated network, but one-layer linear is too simple to get a convincing results.
> > > > > > >
> > > > > > > I would be convinced if the authors could show either for one-layer nonlinear network, or multi-layer linear network.

---

> > > > > > > > ### Author Response · Authors · 2018-12-06
> > > > > > > > **Answer**
> > > > > > > >
> > > > > > > > Q:It is not necessarily to do the real complicated network, but one-layer linear is too simple to get a convincing results.
> > > > > > > > I would be convinced if the authors could show either for one-layer nonlinear network, or multi-layer linear network.
> > > > > > > >
> > > > > > > >
> > > > > > > > A: A multi-layer linear network is actually equivalent in representation capacity to a one-layer network. For instance, consider a network with 2 concat layers, which (using our previous terminology) is given by:
> > > > > > > >
> > > > > > > > (1) W_2 * (W_1 * z + V_1 * one_hot(y) + b_1) + V_2 * one_hot(y) + b_2.
> > > > > > > >
> > > > > > > > Eq. (1) is equivalent to:
> > > > > > > >
> > > > > > > > (W_2 * W_1) * z + (W_2 * V_1 + V_2) * one_hot(y) + (W_2 * b_1 + b_2).
> > > > > > > >
> > > > > > > > By renaming W = W_2 * W_1; V = W_2 * V_1 + V_2 and b = W_2 * b_1 + b_2, then we obtain the same expression as before: W * z + V * one_hot(y) + b.
> > > > > > > >
> > > > > > > > ---
> > > > > > > >
> > > > > > > > One-layer nonlinear network. Consider for example concat and cBN layers and a distribution of one particular neuron with a ReLU nonlinearity.
> > > > > > > > After ReLU, the distribution becomes a Rectified Gaussian (https://en.wikipedia.org/wiki/Rectified_Gaussian_distribution), i.e.,
> > > > > > > > N^R(m_y, s^2) for concat and N^R(m_y, s_y^2) for cBN.
> > > > > > > > The former is characterized by one class-independent and one class-specific parameter, while the latter has 2 class-specific parameters, thus the representation capacity of the latter is higher.

---

### Official Review · AnonReviewer1 · 2018-11-01
**Interesting idea, Convince Results**

**Rating:** 7
**Confidence:** 4

**Review:**

This paper tends to address the instability problem in GAN training by replacing batch normalization(BN) with whitening and coloring transform(WC) to provide a full-feature decorrelation. This paper consider both uncondition and condition cases.
In general, the idea of replacing BN with WC is interesting and well motivated.

The proposed method looks novel to me. Compared with ZCA whitening in Huang et al. 2018, the Cholesky decomposition is much faster and performs better. The experiments show the promising results and demonstrate the proposed method is easily to integrate with other advanced technic. The experimental results also illustrate the role of each components and well supports the motivation of proposed method.

My only concern is that the proposed WC algorithm seems to have capability of applying to many tasks including discriminative scenario. This paper seems to have potential to be a more general paper about the WC method. Why just consider GAN? What is the performance of WC compared with BN/ZCA whiten in other tasks. It would be better if the authors can elaborate the motivation of choosing GAN as the application.

---

> ### Author Response · Authors · 2018-11-26
> **Response to Reviewer #1**
>
> Thank you for your review. Below our answer.
>
> Q: My only concern is that the proposed WC algorithm seems to have capability of applying to many tasks including discriminative scenario. This paper seems to have potential to be a more general paper about the WC method. Why just consider GAN? What is the performance of WC compared with BN/ZCA whiten in other tasks. It would be better if the authors can elaborate the motivation of choosing GAN as the application.
>
> A: Please, note that in the ex-Appendix D (which is now Sec. 6) we actually compare WC with both BN and DBN (proposed in (Huang et al. (2018)) and based on the ZCA whitening) in a discriminative scenario using the protocol suggested in (Huang et al. (2018)). The results reported in the ex-Tab. 10 (now Tab. 7) show that both WC and DBN achieve lower errors than BN. Moreover, WC is only slightly worse than DBN (e.g., using a ResNet-32, WC has a 0.0006 error rate higher than DBN). However, note also that the maximum error rate difference using the protocol suggested in (Huang et al. (2018)) over all the tested normalization techniques, is lower than 0.01. This probably shows that in a discriminative scenario, replacing standardization (BN) with full-feature whitening (e.g., using our WC or DBN) is much less useful than in a GAN scenario, in which we got results drastically different when using WC/cWC with respect to BN/cBN.
> From this empirical analysis we conclude that full-feature whitening shows its major application potential in a GAN setting. The reason behind this different behaviour (marginal accuracy boost in a discriminative scenario vs. large boost in a GAN setting) is probably due to the higher instability of GAN training with respect to discriminative networks. Indeed, as mentioned in Sec. 1, recent papers (Santurkar et al. (2018); Kohler et al. (2018)) show that the main reason of the success of BN relies on an improved training stability. As a consequence, our extension of feature standardization to feature whitening goes in the direction of further improving this stability, which is much more important for GANs than for discriminative networks (please, see also our first answer to Reviewer #3).
> In the new version of the paper we have moved Appendix D to Sec. 6 and we have added a discussion at the end of that section which summarizes the above analysis.
> Finally, note that a second, important motivation of our work, for conditional GANs, is that our cWC can represent class-specific information using more informative filters (Sec. 1), and this second aspect is naturally related to a GAN-based application.

---

### Official Review · AnonReviewer3 · 2018-11-06
**Interesting, but there are some unclear issues**

**Rating:** 7
**Confidence:** 2

**Review:**

This paper proposes to generalize both BN and cBN using Whitening and Coloring based batch normalization. Whitening is an enhanced version of mean subtraction and normalization by standard deviation. Coloring is an enhanced version of per-dimension scaling and shifting.
Evaluation experiments are conducted on different datasets and using different GAN networks and training protocols. Empirical results show improvements over BN and cBN.

The proposed method WC is interesting, but there are some unclear issues.

1. Two motivations for this paper: BN improves the conditioning of the Jacobian, stability of GAN training is related to the conditioning of the Jocobian. These motivate the paper to develop enhanced versions of BN/cBN, as said in the introduction. More discussions why WC can further improve the conditioning over ordinary BN would be better.

2. It is not clear why WC performs better than W_zca C (Table 3), though the improvement is moderate. The difference is that WC uses Cholesky decomposition and ZCA uses eigenvalue decomposition. Compared to W_zca C, WC seems to be an incremental contribution.

3. It is not clear why the proposed method is much faster than ZCA-based whitening.

===========  comments after reading response ===========

The authors make a good response, which clarifies the unclear issues from my first review. I remove the mention of the concurrent submission.

Specially, the new Appendix D with the new Fig. 4 clearly explains and shows the benefit of WC over W_zca.

---

> ### Author Response · Authors · 2018-11-26
> **Response to Reviewer #3 (part 1)**
>
> Thank you for your review. Below our answers.
>
> Q: Two motivations for this paper: BN improves the conditioning of the Jacobian, stability of GAN training is related to the conditioning of the Jacobian. These motivate the paper to develop enhanced versions of BN/cBN, as said in the introduction. More discussions why WC can further improve the conditioning over ordinary BN would be better.
>
> A:  Besides the works mentioned in Sec. 1, different other works showed the relation between the smoothness of the landscape of the loss function and the input-feature normalization and this is the key motivation behind batch-based normalization techniques, including our WC. For instance, ((Huang et al. (2018)), [A],[B]) show that better conditioning of the covariance matrix of the input features leads to better conditioning of the Hessian of the loss function, making the gradient descent weight updates closer to Newton updates. However, BN only performs standardization (of each layer's input). As noticed in (Huang et al. (2018)), when the features are correlated, "standardization barely improves the conditioning of the covariance matrix, whereas whitening remains effective." For example, in 2D, perfectly correlated features "means all points lie close to the line y = x and BN does not change the shape of the distribution" (Huang et al. (2018)).
> Conversely, full-feature whitening completely decorrelates the batch samples, thus potentially improving the smoothness of the loss function. Our empirical results show that this is (significantly) true, at least in a GAN scenario.
> In the new version of the paper we have emphasized the relation between the smoothness of the loss function and the input-feature normalization and the consequent expected advantage in using full-feature whitening in the new Sec. 6 (ex-appendix D, now with a new discussion on this topic at the end of the section).
> Finally, note that a second, fundamental motivation of our work is specific to the conditional GAN setting, where the proposed conditional Coloring can represent richer class-dependent information (see Abstract and Sec. 1, 4, 6).
>
> [A] Y. LeCun et al., "Efficient backprop". In: "Neural Networks: Tricks of the Trade", 1998
> [B] S. Wiesler and H. Ney. "A convergence analysis of log-linear training", NIPS, 2011
>
> ------
>
> Q: It is not clear why WC performs better than W_zca C (Table 3), though the improvement is moderate. The difference is that WC uses Cholesky decomposition and ZCA uses eigenvalue decomposition. Compared to W_zca C, WC seems to be an incremental contribution.
>
> A: Note that W_{zca}C is not the Decorrelated Batch Normalization (DBN) proposed in (Huang et al. (2018)) because, for instance,  no coloring is used in DBN. Hence W_{zca}C is a variant of our WC in which the whitening phase is performed using ZCA. This is explained in the last lines of Sec. 5.1.1.
>
> Concerning the reason why WC performs better than W_{zca}C, we believe this is due to the higher stability of the Cholesky decomposition with respect to the Singular Value Decomposition (SVD) used in the ZCA-whitening. Specifically, in the following we refer to Appendix A.2 of (Huang et al. (2018)), where the backpropagation formulas used for the proposed ZCA-based whitening are presented (we used the same backpropagation in our W_{zca}C experiments). The gradient of the loss with respect to the covariance matrix depends on a matrix called K in Eq. (A.11) of that Appendix. The (i,j)-th element of K is (by definition) inversely proportional to the difference of the (i,j)-th singular values (1/ (sigma_i - sigma_j)) of the covariance matrix. As a consequence, if some of the singular values are identical or very close to each other, then computing K_i,j is ill-conditioned.
> What we empirically observed is that W_{zca}C may be highly unstable and training may start to drastically deteriorate after some iterations. Indeed, the results reported in Tab. 3 refer to the *best* IS-FID values observed during training. After about 40k iterations,
> W_{zca}C suddenly degenerated, collapsing to a model that always produces a constant, uniform grey image. To show this phenomenon, we repeated training of both W_{zca}C and WC and we added a new figure (Fig. 4) in the new paper (please, see the new Appendix D, in which we discuss about W_{zca}C instability issues). The new Fig. 4 shows different  IS/training-iteration curves corresponding to both WC and W_{zca}C. As you can see, the  W_{zca}C training behaviour may drastically degenerate at some point. Conversely, our WC (and cWC) has never showed these drastic instability phenomena.

---

> > ### Author Response · Authors · 2018-11-26
> > **Response to Reviewer #3 (part 2)**
> >
> > Q: It is not clear why the proposed method is much faster than ZCA-based whitening.
> >
> > A: Referring to [C] and using FLOP as the computational measure unit, the Cholesky decomposition, used in our whitening procedure, is 1/3 n^3 FLOPs [C]. Summing the inversion of L (1/3 n^3 FLOPs), we have that our whitening method needs 2/3 n^3 FLOPs. Conversely, SVD, used in the ZCA-based whitening, needs 13 n^3 FLOPs (assuming m = n) [C], which is one order of magnitude higher than our approach.
> >
> > [C] Trefethen and Bau, "Numerical Linear Algebra", 1997
> >
> > -----
> >
> > Q: "Our CIFAR-10 supervised results are higher than all previous works on this dataset." These are somewhat overstated. 8.66 unconditioned IS looks good; however, conditioned IS equal to or better than 9.06 have been reported, e.g. in a concurrent ICLR submission - "Learning Neural Random Fields with Inclusive Auxiliary Generators".
> >
> > A: This remark is not clear to us. Of course we cannot be aware of concurrent ICLR submissions and we think that experimental comparisons at submission time should be done with only already published papers. However, checking  the arXiv version of the paper you mentioned, we noticed that the authors report exactly the same IS we got (IS = 9.06).

---

### Meta-Review · Area_Chair1 · 2018-12-13
**solid idea and results**

**Confidence:** 5
**Recommendation:** Accept (Poster)

**Metareview:**

The paper addresses normalisation and conditioning of GANs. The authors propose to replace class-conditional batch norm with whitening and class-conditional coloring. Evaluation demonstrates that the method performs very well, and the ablation studies confirm the design choices. After extensive discussion, all reviewers agreed that this is a solid contribution, and the paper should be accepted.